# Learnable Fractional Fourier and Graph Fractional Operators for Nonstationary Graph Signals Validated with EEG Seizure Detection

## Abstract

Nonstationary graph signals with time-varying spectral properties and evolving network topologies present fundamental challenges for existing deep learning architectures. We introduce learnable fractional operators that bridge time-frequency analysis and functional connectivity through trainable fractional orders. We propose EEG-GraphFrFT, a unified dual-path framework implementing this approach. The first path employs a fractional Fourier transform with a trainable order to adaptively capture nonstationary, transient patterns. The second path constructs functional networks derived from wPLI and Spectral Granger causality and applies graph fractional operators to model complex network interactions. We establish the minimal theoretical properties required for stable training, namely well-posedness and Hölder-type stability, under mild spectral assumptions. A parameter-efficient low-rank cross interaction integrates the two paths. As a challenging validation, we evaluate on epileptic seizure detection across three public datasets (FMCE, HUP, and Helsinki neonatal EEG) under strict subject-disjoint conditions (no leakage). EEG-GraphFrFT consistently outperforms strong baselines (e.g., EEG-Conformer, Mamba) by approximately 2–8 % in accuracy, while demonstrating robust performance under colored Gaussian noise and channel dropouts, with corresponding improvements in F1 and AUROC. Beyond EEG, the graph fractional operators are task-agnostic and apply broadly to nonstationary graph signals, e.g., traffic sensor networks, climate teleconnections, and multi-asset financial series.

## 1 Introduction

Epilepsy affects over 70 million people worldwide (Thijs et al., 2019). Accurate EEG-based diagnosis is clinically critical yet difficult (Stamoulis et al., 2012): EEG encodes rich spatiotemporal dynamics (Noorlag et al., 2022) and seizure-related state transitions (Li et al., 2019), but expert review of long multi-channel recordings is time-consuming and subjective. Reliable computer-aided diagnosis (CAD) is therefore a central goal (Pontes et al., 2024).Seizure EEG also exemplifies a broader challenge: nonstationary graph signals, where node features and graph topology evolve jointly. Standard deep models often assume stationary features or static graphs; performance degrades when both fail. Similar issues arise in stressed financial markets, disrupted traffic networks, and viral information spread.

Classical Fourier analysis cannot capture time-varying spectra (Yi et al., 2023). Graph Fourier transforms (Shuman et al., 2013) address static networks but not temporal evolution of graph spectra. Advances in fractional calculus and graph signal processing (Guibas et al., 2021a; Ortega et al., 2018) suggest that fractional-order operators are well-suited for such dynamics. AFNO leverages fractional transforms in vision (Guibas et al., 2021a); graph fractional Fourier transforms extend to irregular domains (Alikaşifoğlu et al., 2024). Yet existing approaches typically lack: (i) learnable fractional parameters, (ii) stability guarantees under graph perturbations and noise, and (iii) unified architectures that couple temporal nonstationarity with graph dynamics.

**Related work.** The Fractional Fourier Transform (FrFT) has emerged as a key operator in deep learning due to its unique properties. (Guibas et al., 2021b) developed the Adaptive Fourier Neural Operator (AFNO), which leverages FrFT's rotational characteristics to enable quasi-linear token mixing in vision transformers, reducing computational complexity by 30% compared to standard self-attention. (Liu et al., 2019) combined FrFT with multi-scale wavelet decompositions, creating Fractional Wavelet Scattering Networks for pathological image analysis. (Chen et al., 2025) designed Multi-scale Fractional Fourier Convolutional Neural Networks (MFFCNN) with fractional orders $\alpha$, achieving strong results on segmentation benchmarks. Concurrently, the Graph Fractional Fourier Transform (GFRFT) has extended these concepts to graph structures. (Alikaşifoğlu et al., 2024) established GFRFT as a continuous rotational operator in the vertex-frequency plane. Methodological innovations include fractional translation operators for vertex-localized filtering (Yan et al., 2020) and gradient-based order optimization (Zhang and Li, 2025), with applications demonstrating significant improvements in graph signal denoising (Gan et al., 2023) and classification accuracy (Wei and Yan, 2024).

**Contributions.** We address these gaps with a dual-path framework, instantiated for seizure EEG while remaining applicable to generic nonstationary graph signals. Path 1 performs adaptive time–frequency analysis via a fractional Fourier (FrFT) layer with a learnable order. Path 2 constructs dynamic functional networks using wPLI and Spectral Granger causality, then applies graph fractional operators to model evolving interactions. Under mild spectral assumptions we provide theoretical guarantees—well-posedness and Hölder-type stability—supporting stable training. A parameter-efficient low-rank bilinear fusion integrates the two paths.

On three public datasets (FMCE, HUP, Helsinki-neonatal) with strict subject-disjoint splits, our model consistently improves over strong baselines (EEG-Conformer, FreTS, iTransformer, Brain-JEPA, Mamba, FAPEX) by approximately 2–8 % in accuracy, with corresponding gains in F1 and AUROC, and remains robust under colored noise and channel dropouts. Although we focus on EEG, the proposed learnable fractional operators are task-agnostic and naturally extend to traffic sensor networks, climate teleconnections, and multi-asset financial series. Our contributions can be summarized as follows:

**Learnable fractional operator framework.** A trainable GraphFrFT and fractional operator layer with adaptive exponential modulation for nonstationary time–frequency analysis.

**Graph fractional stability theory.** Hölder-type stability bounds for fractional graph filters under bounded structural perturbations; training is well-posed under mild spectral conditions.

**Unified nonstationary graph modeling.** A dual-path design that fuses signal-space FrFT and graph-space fractional filtering via low-rank fusion, yielding state-of-the-art seizure detection while retaining cross-domain applicability.

## 2 METHOD

The EEG-GraphFrFT framework we proposed is a dual-path architecture designed to comprehensively capture the complex and non-stationary features of electroencephalogram (EEG) signals for seizure detection, as shown in Figure 1. This model processes the original multi-channel EEG input through two complementary and domain-specific channels, each of which is customized to extract different but complementary feature representations. CHAN1 directly operates on time series signals and uses Generalized Fractional Neural Operator (GFNO) with learnable fractional parameters for adaptive time-frequency analysis. CHAN2 builds a dynamic functional connectivity network from the same EEG input, quantifies undirected synchronization (via wPLI) and directed causal effects (via Spectral Granger causality), and then processes it through the graph fractional Fourier Transform (GFRFT) layer to extract the spectral features of the graph. A dedicated feature fusion module integrates the feature mappings of the two paths, and then generates the final prediction through a series of fusion post-processing modules and task-specific classifiers. This design enables the model to synergistically utilize signal space and graph space representations, providing a solid foundation for detecting pathological brain dynamics related to epilepsy.

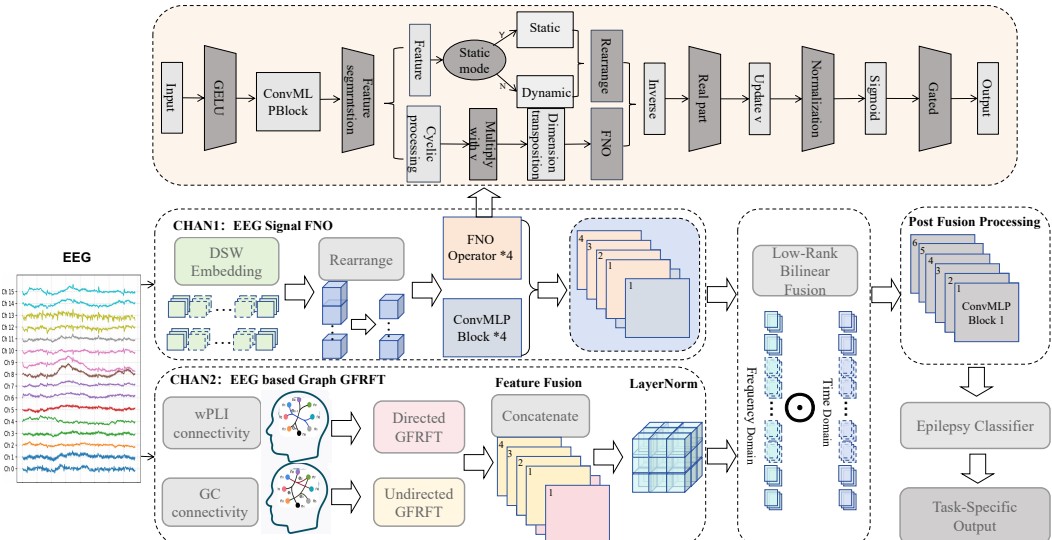

Figure 1: **The EEG-GraphFrFT framework employs a dual-path architecture:** CHAN1 applies trainable fractional neural operators for adaptive time–frequency analysis, while CHAN2 constructs functional networks (wPLI/GC) processed via graph fractional operators. Features are fused via parameter-efficient low-rank bilinear pooling for seizure detection.

## 2.1 GENERALIZED FRACTIONAL NEURAL OPERATOR

Real-world EEG signals are inherently non-stationary, exhibiting complex temporal dynamics such as drifting instantaneous frequencies, intermittent oscillatory patterns, and amplitude-phase modulations, particularly during epileptic seizures. Traditional signal processing techniques, including standard Fourier and wavelet transforms, often fail to adequately capture these behaviors due to their fixed basis functions and inherent assumption of stationarity. To address this limitation, we introduce a *Generalized Fractional Neural Operator (GFNO)*, a novel learnable operator that extends fractional pseudo-differential calculus within a deep learning framework, enabling adaptive time-frequency analysis tailored to non-stationary EEG characteristics. The mathematical foundation of our approach rests on the generalization of classical convolution and differentiation through fractional pseudo-differential operators. These operators enable adaptive joint time-frequency analysis by incorporating a rotational parameter that extends traditional Fourier analysis, enabling more flexible representation of non-stationary, highly variable signals such as EEG.

**Definition 1** (Fractional Pseudo-Differential Operator). Let $a(t, \xi)$ be a sufficiently smooth symbol function defined on $\mathbb{R}_t \times \mathbb{R}_\xi$. For any rotation angle $\theta \notin \pi\mathbb{Z}$ and for any $\phi \in L^2(\mathbb{R})$, the associated *fractional pseudo-differential operator* $T_a^\theta : L^2(\mathbb{R}) \to L^2(\mathbb{R})$ is defined as:

$$(T_a^\theta \phi)(t) \ = \ \int_{\mathbb{R}} K_{-\theta}(t, \xi) \, a(t, \xi) \, \widehat{\phi}^\theta(\xi) \, d\xi, \tag{1}$$

where $\widehat{\phi}^\theta(\xi) \equiv \mathcal{F}^\theta[\phi](\xi)$ denotes the Fractional Fourier Transform (FrFT) of $\phi$ of order $\theta$, and $K_{-\theta}(t, \xi)$ represents the kernel associated with the fractional transformation.

To operationalize this mathematical framework within a deep learning architecture, we introduce the *Neural Fractional Operator (NFO)*, which parameterizes the symbol function $a(t, \xi)$ using learnable components while maintaining the theoretical properties of fractional operators. Let $X \in \mathbb{R}^{L \times C}$ represent a multivariate EEG signal with $L$ time samples and $C$ channels. The symbol function $a(t, \xi)$ is factorized into separable time-dependent and frequency-dependent components $a(t, \xi) = u(t)v(\xi)$, where $u : \mathbb{R} \to \mathbb{R}^C$ and $v : \mathbb{R} \to \mathbb{R}^C$ are learned functions capturing temporal and spectral profiles respectively. The Neural Fractional Operator $\mathcal{T}_a^\theta : L^2(\mathbb{R})^C \to L^2(\mathbb{R})^C$ of order $\theta \notin \pi\mathbb{Z}$ operates on signal $\varphi \in L^2(\mathbb{R})^C$ as:

$$\left(\mathcal{T}_a^\theta \varphi\right)(t) = u(t)\mathcal{F}^{-\theta}\left[v(\xi) \odot \mathcal{F}^\theta[\varphi](\xi)\right](t) \tag{2}$$

where $F^\theta[\cdot]$ denotes the FrFT of order $\theta$, and $\odot$ is channel-wise multiplication. The NFO is parameterized by two hypernets $\text{HyperNet}^t$ and $\text{HyperNet}^\xi$ producing $u(t)$ and $v(\xi)$. We set the trainable order as $\theta = \frac{\pi}{2}\tanh(\beta_\theta)$ with learnable $\beta_\theta$ (compact domain, stable gradients). Both hypernets are single-layer MLPs with Snake activation $A_\beta(x) = x + \beta^{-1}\sin^2(\beta x)$ and learnable $\beta > 0$ to mitigate spectral bias. To stabilize high-frequency behavior we apply a nonnegative frequency gate $w(\xi) = \text{SoftPlus}(\gamma_0 + \gamma_1|\xi|)$ and use $\tilde{v}(\xi) = \exp(-w(\xi))\,v(\xi)$ inside the operator; this yields monotone, learnable attenuation without numerical instability.

**Positional encoding.** Positional information is encoded via Fourier embeddings:

$$z(t) = \big[t_{\text{rescaled}},\ \{\cos(f_k \cdot w)\}_{k=1}^B,\ \{\sin(f_k \cdot w)\}_{k=1}^B\big],$$

where $t_{\text{rescaled}} = t/T$, $w = 2\pi t/T$, and $f_k = 2^{k-1}$ for $k = 1,\ldots,B$. Here $T$ is the sequence length and $\{f_k\}$ form geometric frequency bands, yielding $z(t) \in \mathbb{R}^{1+2B}$.

## 2.2 Graph-Based Functional Connectivity Analysis with Fractional Fourier Transform

**Functional Connectivity Networks with wPLI and GC** We have established a functional connection network. It captures the dynamic changes among complex regions in the brain activity of epilepsy patients. We use two complementary metrics. The Weighted Phase Lag Index (wPLI) measures phase-based synchronization (see Appendix B.2.1). Granger causality (GC) captures the directed causal information flow in the frequency domain (see Appendix B.2.2). The combination of this dual-function network construction method provides a complete capture of brain network information. It integrates undirected functional coupling with causal effects during epileptic seizures. See Appendix B.2.3 for details on construction of wPLI and GC graphs from raw signals. *Notation:* we use $(\cdot)^\top$ for transpose and $(\cdot)^H$ for conjugate transpose; for real-symmetric $L_{\text{sym}}$ we write $L_{\text{sym}} = U\Lambda U^\top$, while for Hermitian $L_H$ we write $L_H = U\Lambda U^H$.

**Graph Fractional Operators via Functional Calculus (Unified)** We operate *exclusively* on Hermitian (PSD) Laplacians to ensure real spectra and stable gradients. Let $\mathbf{L}_\star$ denote the graph operator used for the spectral calculus: $\mathbf{L}_\star = \mathbf{L}_{\text{sym}}$ for undirected wPLI graphs, and $\mathbf{L}_\star = \mathbf{L}_H$ for GC graphs (Hermitianized; Appendix B.2.5). With $\mathbf{L}_\star = \mathbf{U}\boldsymbol{\Lambda}\mathbf{U}^H$ and $\alpha \in (0,1]$, we define the fractional family

$$\mathbf{U}_{\theta,\alpha}(\mathbf{L}_\star) = e^{-i\theta\mathbf{L}_\star^\alpha}, \qquad \mathbf{H}_{\tau,\alpha}(\mathbf{L}_\star) = e^{-\tau\mathbf{L}_\star^\alpha},$$

and use their real/imaginary parts as learnable graph-spectral filters. Gradients are analytic: $\partial_\alpha\boldsymbol{\Lambda}^\alpha = \boldsymbol{\Lambda}^\alpha \circ \log\boldsymbol{\Lambda}$ and $\partial_\theta e^{-i\theta\boldsymbol{\Lambda}^\alpha} = -i\,\boldsymbol{\Lambda}^\alpha e^{-i\theta\boldsymbol{\Lambda}^\alpha}$.

**Implementation & complexity.** During training we approximate $\mathbf{L}_\star^\alpha$ and the matrix exponentials by Chebyshev/Lanczos expansions of order $K$ (typically $K = 8\text{–}16$), giving $\mathcal{O}(K|E|)$ time and $\mathcal{O}(K|V|)$ memory per layer.

**Directed connectivity.** For GC-derived graphs we build $\mathbf{L}_H = \mathbf{D}_s - \boldsymbol{\Gamma}\odot\mathbf{W}_s$ (unit-modulus phase $\boldsymbol{\Gamma}$; Appendix B.2.2–B.2.3), which is Hermitian PSD. All fractional operators are applied to $\mathbf{L}_H$ to guarantee a real spectrum and numerical stability.

**Extended Fractional Graph Calculus** The fractional graph derivative of order $\beta$ is defined as $\mathcal{D}^\beta = F_G^{-\beta}\Lambda_\beta F_G^\beta$, where $\Lambda_\beta = \text{diag}((\lambda_k)^\beta)$ and $\lambda_k$ are graph frequencies, extending classical fractional calculus to irregular domains. For a symmetrized adjacency matrix $W_s$ and unit-modulus phase matrix $\Gamma_q = (\Upsilon_q(i,j))$ satisfying $\Upsilon_q(i,j) = \overline{\Upsilon_q(j,i)}$ for all $i,j$, define $L_H = D_s - \Gamma_q \odot W_s$, where $D_s = \text{diag}(W_s\mathbf{1})$. Then $L_H = L_H^H$ and $x^H L_H x = \frac{1}{2}\sum_{i,j} W_{s,ij}\,|x_i - \Upsilon_q(i,j)x_j|^2 \geq 0$. Then, $L_H$ is Hermitian ($L_H = L_H^*$), positive semidefinite ($x^* L_H x \geq 0$ for all $x \in \mathbb{C}^n$), recovers the standard symmetric Laplacian $L_{\text{sym}}$ in the undirected case ($w_{ij} = w_{ji}$, $\Gamma_q \equiv 1$), and has eigenvalues on the nonnegative real axis by Gershgorin's theorem (proof in Appendix B.2.6). For any diagonalization $L_H = UVU^*$ with $V = \text{diag}(\nu_k)$, define $\log V = \text{diag}(\log|\nu_k| + j\arg(\nu_k))$ and $\nu_k^\alpha = \exp(\alpha\log\nu_k)$, using the principal branch $\arg(\nu_k) \in (-\pi,\pi]$. Zero eigenvalues are shifted by a small $\varepsilon$, and tiny negative parts from numerical roundoff are clamped to zero, ensuring $\log V$ and $V^\alpha$ are well-defined and numerically stable.

**GFRFT-Based Feature Extraction** To extract discriminative features from functional connectivity networks, a dual-mode Graph Fourier Transform (GFT) framework is implemented, adapting to undirected (wPLI-based) and directed (GC-based) graph structures. The core innovation is an adaptive eigendecomposition strategy, based on graph directionality and a learnable fractional order parameter $\alpha$.

**Undirected Graph Processing (wPLI):** For undirected networks derived from wPLI, the symmetric normalized Laplacian is computed as $\mathbf{L}_{\text{sym}} = \mathbf{I} - \mathbf{D}^{-1/2}\mathbf{A}\mathbf{D}^{-1/2}$, where $\mathbf{D}$ is the degree matrix and $\mathbf{A}$ is the symmetric adjacency matrix. Eigendecomposition preserves the spectral gap: $\mathbf{L}_{\text{sym}} = \mathbf{U}\boldsymbol{\Lambda}\mathbf{U}^{\top} = \sum_{k=0}^{N-1} \lambda_k \mathbf{u}_k \mathbf{u}_k^{\top}$, with eigenvalues ordered $0 = \lambda_0 \leq \lambda_1 \leq \cdots \leq \lambda_{N-1} = \lambda_{\max}$. The learnable GFRFT is applied as $\mathbf{Z}_{\text{undir}} = \mathbf{U}\boldsymbol{\Lambda}^{\alpha}\mathbf{U}^{\top}\mathbf{X} = \sum_{k=0}^{N-1} \lambda_k^{\alpha}(\mathbf{u}_k^{\top}\mathbf{X})\mathbf{u}_k$, where $\alpha$ is initialized at 0.5 and optimized via $\nabla_{\alpha}\mathcal{L} = \text{Tr}\left(\frac{\partial\mathcal{L}}{\partial\mathbf{Z}_{\text{undir}}}^{\top}\mathbf{U}(\boldsymbol{\Lambda}^{\alpha} \circ \log\boldsymbol{\Lambda})\mathbf{U}^{\top}\mathbf{X}\right)$.

**Directed graphs (GC) via Hermitianization.** For GC-derived directed connectivity we use the Hermitian Laplacian $L_H = D_s - \Gamma \odot W_s$ (unit-modulus phase $\Gamma$; Appendix B.2.2–B.2.3). All fractional operators are applied to $L_H$, ensuring a real PSD spectrum. For GC-derived directed connectivity we apply fractional operators to $L_H$ *per frequency band* (each band is Hermitianized and diagonalized independently); adjacent-band orders $\{\alpha_b\}$ are tied via a total-variation penalty to reduce overfitting.

**Unified processing.** We *always* apply fractional operators on a Hermitian PSD Laplacian:

$$\mathbf{Z} = \mathbf{U}\,\boldsymbol{\Lambda}^{\alpha}\,\mathbf{U}^{H}\,\mathbf{X}, \quad \mathbf{L}_{\star} = \begin{cases} \mathbf{L}_{\text{sym}} & \text{(wPLI, undirected),} \\ \mathbf{L}_{H} & \text{(GC, directed; Hermitianized)} . \end{cases}$$

This removes the unstable $\mathbf{A} = \mathbf{U}\mathbf{V}\mathbf{U}^{-1}$ branch and keeps gradients real-valued.

**Multi-band Processing for GC:** For Granger Causality networks, each frequency band is processed separately. GC matrices are computed for bands $\mathbf{A}_{\delta}, \mathbf{A}_{\theta}, \mathbf{A}_{\alpha}, \mathbf{A}_{\beta}, \mathbf{A}_{\gamma}$, and directed GFRFT is applied with independent $\alpha$ parameters. Band-specific features $\mathbf{h}_{\delta}, \mathbf{h}_{\theta}, \mathbf{h}_{\alpha}, \mathbf{h}_{\beta}, \mathbf{h}_{\gamma}$ are extracted and concatenated: $\mathbf{h}_{\text{gc}} = [\mathbf{h}_{\delta} \oplus \mathbf{h}_{\theta} \oplus \mathbf{h}_{\alpha} \oplus \mathbf{h}_{\beta} \oplus \mathbf{h}_{\gamma}]$. We share $\alpha$ across adjacent bands with a total-variation penalty on $\{\alpha_b\}$ to reduce overfitting.

**Feature Fusion:** The final representation combines modalities: $\mathbf{h}_{\text{final}} = \mathbf{h}_{\text{eeg}} \oplus \mathbf{h}_{\text{wpli}} \oplus \mathbf{h}_{\text{gc}}$, where $\mathbf{h}_{\text{eeg}}$ are features from raw EEG, $\mathbf{h}_{\text{wpli}}$ from wPLI graphs, and $\mathbf{h}_{\text{gc}}$ from GC graphs.

**Feature Fusion via Low-Rank Bilinear Interaction** To integrate time-frequency (CHAN1) and graph (CHAN2) features, a low-rank bilinear fusion mechanism captures multiplicative interactions between feature vectors $\mathbf{s} \in \mathbb{R}^{d_s}$ and $\mathbf{g} \in \mathbb{R}^{d_g}$. This approach is more expressive than concatenation or addition and is parameter-efficient due to low-rank constraints. The output $\mathbf{y} \in \mathbb{R}^{d_o}$ is computed as:

$$\mathbf{y} = \mathbf{U}^{\top}\left[(\mathbf{V}_s^{\top}\mathbf{s}) \odot (\mathbf{V}_g^{\top}\mathbf{g})\right] + \mathbf{b}, \tag{3}$$

where $\mathbf{V}_s \in \mathbb{R}^{d_s \times r}$, $\mathbf{V}_g \in \mathbb{R}^{d_g \times r}$ project features to a shared $r$-dimensional space ($r \ll \min(d_s, d_g)$), $\mathbf{U} \in \mathbb{R}^{r \times d_o}$ projects to the output space, and $\odot$ denotes the Hadamard product. This models second-order interactions efficiently, with details and derivations provided in Appendix B.2.9.

**Stability of Graph Fractional Filters under Perturbations** We study the stability of fractional graph filters under structural perturbations of the Laplacian. Throughout this subsection we assume that $L, \tilde{L} \in \mathbb{R}^{n \times n}$ are real symmetric positive semidefinite (PSD) Laplacians with spectra in $[0, \Lambda_{\max}]$, and that $\alpha \in (0, 1]$, $\tau > 0$. We write $\Delta = \tilde{L} - L$ and use $\|\cdot\|_2$ for the operator norm. Fractional powers are defined via functional calculus: $L^{\alpha} = U\Lambda^{\alpha}U^{\top}$ for $L = U\Lambda U^{\top}$. All proofs are deferred to Appendix B.2.7.

**Theorem 1** (Hölder stability of fractional graph filters). *Let $L, \tilde{L} \succeq 0$ be PSD Laplacians, $\alpha \in (0, 1]$, and $\tau > 0$. Then there exists a constant $C_{\alpha} > 0$ depending only on $\alpha$ and a priori bounds on $\|L\|_2, \|\tilde{L}\|_2$ such that*

$$\left\|e^{-\tau L^{\alpha}} - e^{-\tau \tilde{L}^{\alpha}}\right\|_2 \leq \tau\, C_{\alpha}\, \|L - \tilde{L}\|_2^{\min\{1,\alpha\}}. \tag{4}$$

*Proof sketch. First, fractional powers are Hölder continuous on PSD cones:* $\|L^\alpha - \tilde{L}^\alpha\|_2 \leq C_\alpha \|L - \tilde{L}\|_2^{\min\{1,\alpha\}}$ *for* $\alpha \in (0, 1]$. *Second, use the integral representation* $e^{-A} - e^{-B} = \int_0^1 e^{-(1-s)A} (B - A) e^{-sB} \, ds$ *with* $A = \tau L^\alpha$, $B = \tau \tilde{L}^\alpha$; *contractions* $\|e^{-X}\|_2 \leq 1$ *for* $X \succeq 0$ *give* $\|e^{-\tau L^\alpha} - e^{-\tau \tilde{L}^\alpha}\|_2 \leq \tau \|L^\alpha - \tilde{L}^\alpha\|_2$. *Combine the two bounds.* □

**Theorem 2** (High-frequency attenuation on spectral subspaces). *Let* $P_E = \mathbf{1}_{[\lambda_0, \infty)}(L)$ *be the spectral projector of* $L$ *onto eigenvalues* $\geq \lambda_0 > 0$. *If* $\|\tilde{L} - L\|_2 \leq \varepsilon$ *with* $\varepsilon \in [0, \lambda_0)$, *then*

$$\left\| P_E \left( e^{-\tau L^\alpha} - e^{-\tau \tilde{L}^\alpha} \right) P_E \right\|_2 \leq \tau \, e^{-\tau \, ((\lambda_0 - \varepsilon)_+)^\alpha} \, \|L^\alpha - \tilde{L}^\alpha\|_2 \leq \tau \, C_\alpha \, e^{-\tau \, ((\lambda_0 - \varepsilon)_+)^\alpha} \, \|L - \tilde{L}\|_2^{\min\{1,\alpha\}}. \tag{5}$$

*where* $(\cdot)_+ := \max(0, x)$. *Proof sketch. On the subspace* $E$, *the spectrum of* $L^\alpha$ *lies in* $[\lambda_0^\alpha, \infty)$ *while Weyl's inequality yields a shift at most* $\varepsilon$ *for* $\tilde{L}$. *Hence* $\|e^{-\tau L^\alpha}|_E\|_2 \leq e^{-\tau \lambda_0^\alpha}$ *and* $\|e^{-\tau \tilde{L}^\alpha}|_E\|_2 \leq e^{-\tau \, ((\lambda_0 - \varepsilon)_+)^\alpha}$. *Apply the integral identity as in Theorem 1 restricted to* $E$ *to obtain the exponential factor.* □

**Implications.** The above bounds imply sublinear sensitivity to structural noise for $\alpha < 1$, and nonexpansiveness of heat-type filters ($\|e^{-\tau L^\alpha}\|_2 \leq 1$). Combined with our empirical noise and dropout robustness, they provide a theory-to-practice explanation for the gains of fractional operators in seizure detection. Proofs are in Appendix B.2.7.

## 3 EXPERIMENT

### 3.1 DATA PREPROCESSING

Neuroelectrical signals are prone to noise and artifacts that obscure neurodynamic features, particularly in high-frequency bands critical for epilepsy research. To preserve data integrity and physiological significance, we applied a strict preprocessing pipeline using MNE v1.9.0 and EEGLAB v2025.0. As summarized in Table 1, all EEG records were resampled to 512 Hz. A 1 Hz bandwidth notch filter removed 50/60 Hz line noise and harmonics, followed by a 0.1–200 Hz band-pass filter to capture relevant subbands ($\delta$: 0.5–4 Hz, $\theta$: 4–8 Hz, $\alpha$: 8–12 Hz, $\beta$: 13–25 Hz, low $\gamma$: 26–80 Hz, HFOs: >80 Hz). Artifact subspace reconstruction (ASR; burst criterion: 10 SD) mitigated high-pass filtering artifacts. Independent component analysis (ICA) using the Picard algorithm identified and removed ocular, muscular, and cardiac artifacts through combined manual inspection and ICLabel automation. Common average referencing (CAR) was applied unless pre-referenced. Training parameters are provided in Table A3.

Table 1: All datasets were standardized to 64 channels based on a common electrode layout. When the original number of channels was less than 64, spatial interpolation was applied for upsampling. For recordings with more than 64 channels, electrodes were first selected according to the standard 10-20 system; if the number of standard channels still exceeded the target maximum, the top 64 channels were retained based on energy density. Then all data were sectioned into 4 seconds long sliding window slices of 0.5 seconds.

| Dataset | Confidentiality | Species | # Subj. | Modality | # Ch. | # Samples | Duration | Stride |
|---|---|---|---|---|---|---|---|---|
| FMCE (Li et al., 2021) | Public | Human | 65 | ECoG/SEEG | 52-232 | 34,320 | 4 s | 0.5 s |
| HUP (Kini et al., 2019) | Public | Human | 73 | ECoG/SEEG | 47-216 | 182,937 | 4 s | 0.5 s |
| Helsinki Neonatal EEG (Stevenson et al., 2019) | Public | Human | 79 | EEG | 21 | 805,097 | 4 s | 0.5 s |

### 3.2 RESULTS AND ANALYSIS

Our proposed EEG-GraphFrFT framework demonstrates superior and robust performance across all evaluation metrics and datasets, achieves the best performance among evaluated baselines for automated seizure detection. The comprehensive analysis below details its effectiveness, robustness, and the contribution of its core components.

**Overall Performance** EEG-GraphFrFT achieves strong performance across all datasets (Table 2), with accuracy scores of **94.12%** (HUP), **94.54%** (FMCE), and **97.44%** (Helsinki), substantially

outperforming second-best models—FAPEX (93.39% on HUP) and Mamba (91.82% on FMCE, 95.25% on Helsinki). The model demonstrates consistent superiority across all metrics, achieving the highest **F1 scores** and **AUC** values (0.9756 on FMCE vs. 0.9578 for EEG-Conformer and 0.9563 for Mamba), indicating exceptional precision-recall balance and ranking capability. Crucially, EEG-GraphFrFT exhibits outstanding sensitivity (**97.21%** on HUP) and specificity (**98.99%** on Helsinki), demonstrating equal proficiency in seizure detection and non-seizure rejection—critical for clinical deployment to minimize false alarms and missed detections. The performance gap highlights the necessity of domain-specific designs: the iTransformer lags considerably (85.87% vs. 94.12% on HUP), while frequency-based FreTS is surpassed by our integration of *learnable* time-frequency representations with graph-based connectivity analysis, outperforming fixed Fourier transforms.

Table 2: Performance comparison of different models on epileptic EEG classification tasks. The **best** results are highlighted in **red** and the **second-best** results are highlighted in **blue** for each metric within each dataset.

| Models | HUP | | | | | FMCE | | | | | Helsinki Neonatal EEG | | | | |
|---|---|---|---|---|---|---|---|---|---|---|---|---|---|---|---|
| Metrics | Acc | F1 | AUC | Sens | Spec | Acc | F1 | AUC | Sens | Spec | Acc | F1 | AUC | Sens | Spec |
| EEG-Conformer (Song et al., 2023) | 0.9026 | 0.9012 | 0.9259 | 0.9055 | 0.9102 | 0.8990 | 0.8990 | 0.9578 | 0.8807 | 0.9100 | 0.8843 | 0.8935 | 0.9095 | 0.9177 | 0.9134 |
| Jepa (Dong et al., 2024) | 0.7917 | 0.7655 | 0.8225 | 0.7884 | 0.8139 | 0.8691 | 0.8376 | 0.8578 | 0.8626 | 0.8898 | 0.8431 | 0.8628 | 0.8728 | 0.8333 | 0.8571 |
| Mamba (Gui et al., 2024) | 0.9107 | 0.9115 | 0.9485 | 0.9347 | 0.9308 | 0.9182 | 0.9183 | 0.9563 | 0.9533 | 0.9246 | 0.9525 | 0.9549 | 0.9839 | 0.9568 | 0.9644 |
| iTransformer (Liu et al., 2023) | 0.8587 | 0.8580 | 0.8717 | 0.8617 | 0.9033 | 0.8578 | 0.8574 | 0.9235 | 0.7896 | 0.8835 | 0.8725 | 0.8903 | 0.9357 | 0.9115 | 0.8639 |
| FreTS (Yi et al., 2023) | 0.8151 | 0.8092 | 0.8003 | 0.8367 | 0.8865 | 0.8460 | 0.8455 | 0.9104 | 0.8367 | 0.8865 | 0.9076 | 0.9087 | 0.9054 | 0.9522 | 0.9451 |
| FAPEX (Zheng et al., 2025) | 0.9339 | 0.9339 | 0.9692 | 0.9364 | 0.9728 | 0.9005 | 0.9003 | 0.9567 | 0.9341 | 0.9588 | 0.9404 | 0.9449 | 0.9816 | 0.9406 | 0.9611 |
| **EEG-GraphFrFT** | **0.9412** | **0.9406** | **0.9750** | **0.9721** | **0.9683** | **0.9454** | **0.9412** | **0.9756** | **0.9528** | **0.9746** | **0.9744** | **0.9739** | **0.9831** | **0.9608** | **0.9899** |

**Protocol fairness.** All baselines are tuned under the same subject-disjoint protocol and early-stopping rule; when authors recommend default configs (e.g., Brain-JEPA), we adopt them verbatim. To ensure the fairness of the experimental setup, a comprehensive comparison with additional baseline models is provided in Table A2.

**Exceptional Noise Robustness** EEG-GraphFrFT exhibits exceptional noise robustness in additive pink noise tests (Table 3), maintaining **93.76%** accuracy under maximum noise (SD=1)—a mere **0.8%** decrease from clean data versus **6.8%** (Mamba) and **4.0%** (EEG-Conformer) drops. F1 score decreases only **0.2%** compared to **7.1%** (Mamba) and **3.9%** (EEG-Conformer), highlighting superior stability. For complete noise robustness experimental data, refer to Table A1; for the visualization of EEG data with superimposed noise, see FigureA1; and for the robustness results of various models under different noise intensities for each metric, see FigureA2.

This robustness originates from our fractional operators: CHAN1's learnable FrFT adaptively filters noise via discriminative fractional orders, while CHAN2's GFRFT processes inherently stable functional connections. The gated fusion mechanism leverages the Hölder stability and exponential high-frequency attenuation properties of fractional graph operators to dynamically integrate noise-robust features from the fractional channels (Section 2.2), in stark contrast to EEG-Conformer's self-attention that tends to overfit to noise patterns and Mamba's state-space model that exhibits heightened sensitivity to perturbations due to the lack of such structural stability guarantees.

Table 3: Performance comparison of different models on epileptic EEG classification tasks under noise conditions for the FMCE dataset. For Pink Noise conditions, the **best** and **second-best** percentage changes (indicating noise robustness) are highlighted for each metric.

| Models | Original | | | | | Pink Noise (SD=1) | | | | |
|---|---|---|---|---|---|---|---|---|---|---|
| Metrics | Acc | F1 | AUC | Sens | Spec | Acc | F1 | AUC | Sens | Spec |
| EEG-Conformer (Song et al., 2023) | 0.8990 | 0.8990 | 0.9578 | 0.8807 | 0.9100 | 0.8630 (-4.0%) | 0.8636 (-3.9%) | 0.8662 (-9.6%) | 0.8240 (-6.4%) | 0.8910 (-2.1%) |
| Mamba (Gui et al., 2024) | 0.9182 | 0.9183 | 0.9568 | 0.9533 | 0.9246 | 0.8559 (-6.8%) | 0.8533 (-7.1%) | 0.9337 (-2.4%) | 0.8400 (-11.9%) | 0.8700 (-5.9%) |
| FAPEX (Zheng et al., 2025) | 0.9005 | 0.9003 | 0.9567 | 0.9341 | 0.9588 | 0.8899 (-1.2%) | 0.8974 (-0.3%) | 0.8976 (-6.2%) | 0.8841 (-5.4%) | 0.9388 (-2.1%) |
| **EEG-GraphFrFT** | **0.9454** | **0.9412** | **0.9756** | **0.9528** | **0.9746** | **0.9376 (-0.8%)** | **0.9393 (-0.2%)** | **0.9706 (-0.5%)** | **0.9378 (-1.6%)** | **0.9696 (-0.5%)** |

**Ablation Study and Component Analysis** Ablation results (Table 4) reveal critical insights: 1) **Learnable $\alpha$ is fundamental**: Optimal configuration (both channels with trained $\alpha$) achieves 94.5% accuracy, while fixed $\alpha = 1$ (standard FT) causes significant 5.0% drop (89.9%), confirming adaptivity's importance for capturing non-stationary dynamics; 2) **Graph pathway synergy**: Removing CHAN2 causes 3.3% performance drop (91.4%), proving functional connectivity provides unique complementary information. Combined removal of CHAN2 and CHAN1 adaptivity collapses performance to 69.1%; 3) **Adaptive spectral filtering**: Learned $\alpha$ shows domain-specific optimiza-

tion—CHAN1 averages $\approx 0.5$ (complete distribution of alpha values in 4 Layers is provided in Figure A3), while CHAN2 tailors the fractional order parameter to different graph types, with $\alpha$ values of 0.4926 for wPLI and 1.0065 for GC (complete distribution of alpha values in 3 Layers is provided in Figure A4). The concentration of CHAN1's FNO average $\alpha$ near 0.5 achieves an optimal time–frequency balance for non-stationary EEG transients. In CHAN2, GFRFT's $\alpha$ is 0.4926 for wPLI (undirected), maintaining a similar balance for synchronization analysis, while its $\alpha$ of 1.0065 for GC (directed) favors a pure spectral approach to model causal flow. This demonstrates the model's adaptive learning of domain-specific fractional orders. Meanwhile, Results in Table 4 (row 4) reveal a critical divergence: the fixed-order transform ($\alpha=1$) increases AUC by 0.4% yet degrades essential clinical classification metrics. This shows the advantages of our learnable fractional order method in diagnostic indicators, thereby ensuring its application value in clinical practice.

Table 4: Performance of EEG-GraphFrFT model on FMCE dataset under different channel settings and fractional order ($\alpha$) values. Percentage changes relative to the first row are shown in parentheses.

| Channel Settings | Acc | F1 | AUC | Sens | Spec |
|---|---|---|---|---|---|
| **CHAN1: trained $\alpha$, CHAN2: trained $\alpha$** | **0.9454** | **0.9412** | **0.9756** | **0.9528** | **0.9746** |
| CHAN1: trained $\alpha$, CHAN2: $\alpha = 1$ | 0.9378 (-0.8%) | 0.9377 (-0.4%) | 0.9759 (+0.0%) | 0.9368 (-1.7%) | 0.9698 (-0.5%) |
| CHAN1: $\alpha = 1$, CHAN2: $\alpha = 1$ | 0.8986 (-5.0%) | 0.8991 (-4.5%) | 0.9795 (+0.4%) | 0.8913 (-6.5%) | 0.9120 (-6.4%) |
| CHAN1: $\alpha = 1.5$, CHAN2: $\alpha = 1.5$ | 0.9213 (-2.5%) | 0.9210 (-2.1%) | 0.9595 (-1.7%) | 0.8902 (-6.6%) | 0.9426 (-3.3%) |
| Remove CHAN2 (only CHAN1: trained $\alpha$) | 0.9138 (-3.3%) | 0.9199 (-2.3%) | 0.9558 (-2.0%) | 0.9062 (-4.9%) | 0.9494 (-2.6%) |
| Remove CHAN2 (only CHAN1: $\alpha = 1$) | 0.6910 (-26.9%) | 0.6984 (-25.8%) | 0.7223 (-26.0%) | 0.6371 (-33.1%) | 0.7367 (-24.4%) |

The exceptional performance and robustness of our EEG-GraphFrFT model are fundamentally attributed to its capacity to extract such highly discriminative features from the input signals. As demonstrated in Figure 2, the functional connectivity networks constructed by our framework successfully capture the stark contrast between interictal and ictal brain states. The wPLI analysis reveals a dramatic emergence of strong, widespread phase-synchronization across the brain network during a seizure—a hallmark of epileptic activity. It is precisely this characteristic *hypersynchronization* that our dual-path architecture, particularly the graph fractional Fourier transform in CHAN2, is designed to detect and characterize in an adaptive manner. The quantitative results from the ablation study (Table 4), which show a significant performance drop when the graph pathway (CHAN2) is removed, provide direct empirical evidence that leveraging this graph-structured information is crucial for the model's success. Furthermore, analysis of specific channels (e.g., O1 and O2 in occipital lobe, T5 and T6 in posterior temporal lobe, Pz, P3, and P4 in parietal lobe) during seizures reveals enhanced synchronization and pathological activities in these regions, which our model effectively captures through GFRFT, demonstrating its sensitivity to localized epileptic networks.

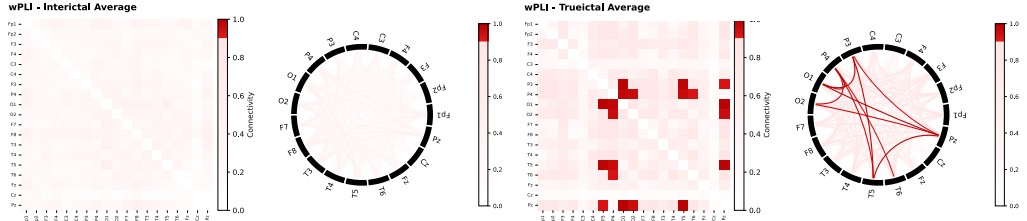

Figure 2: **Group-averaged functional connectivity using wPLI.** Left: Interictal state shows sparse and weak synchronization. Right: Ictal state exhibits dense hypersynchronization. Insets report two graph-spectral summaries *after* the fractional graph operator: the cumulative energy curve $E_{\text{cum}}(k) = \sum_{i=0}^{k} \|\hat{x}_i\|^2$ over Laplacian eigenmodes (ordered by $\lambda$), and the low-frequency energy ratio $R_{\text{low}} = \left( \sum_{\lambda_i \leq \lambda_0} \|\hat{x}_i\|^2 \right) / \left( \sum_i \|\hat{x}_i\|^2 \right)$. Ictal curves rise faster and yield higher $R_{\text{low}}$ together with lower spectral entropy $H_g = -\sum_i p_i \log p_i$ (with $p_i \propto \|\hat{x}_i\|^2$), indicating that the fractional operator concentrates energy on large-scale coherent modes consistent with seizure dynamics. Additional wPLI and GC connectivity patterns for 10 samples are shown in Figure A5 and Figure A6, respectively. Full group-averaged graphs including GC analysis are provided in Figure A7.

Our designed GFRFT demonstrates exceptional feature discrimination capabilities when processing both directed Granger Causality (GC) and undirected wPLI networks. Taking GC as an example,

this advantage is further validated in Figure 3 (the complete results including wPLI are shown in Figure A8). The model transforms the raw, asymmetric GC input matrix which captures directed causal influences between brain regions—through three adaptive GFRFT layers. The key innovation lies in a mathematical transformation that shifts the analytical perspective from simple connection strength (the input view) to the extraction of significant dynamic network patterns, which are encoded in the magnitude and phase of complex-valued features. The ictal (seizure) state, characterized by a strong pathological causal network, activates the GFRFT filters to produce a sparse output of high-magnitude components (see Figure 3 b-d), effectively concentrating the signature of the epileptic seizure. In contrast, the interictal state yields a lower-energy, more diffuse response (see Figure 3 g-i). This demonstrates that our GFRFT not only preserves the input structure but, more importantly, actively refines and distills the complex directed brain dynamics into a potent and highly separable feature representation, which is ultimately projected by the linear layer into a clear and classifiable signature for seizure detection (see Figure 3 e,j). The increased concentration of energy in low-order graph modes, together with stronger phase consistency, is consistent with clinical observations of hypersynchronization and network densification during ictal periods.

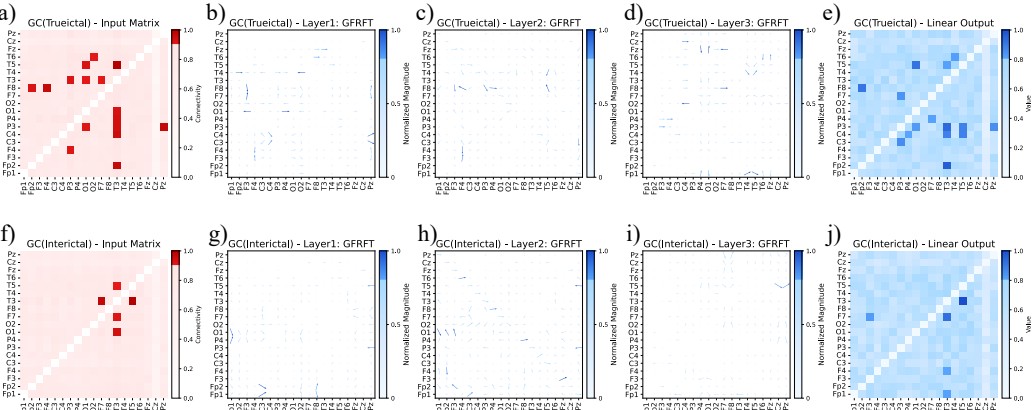

Figure 3: **Processing pipeline for Granger-causality (GC) connectivity through three GFRFT layers.** Panels (a–e) show an ictal trial and panels (f–j) an interictal trial. Arrow length encodes magnitude and arrow angle encodes phase of complex graph-spectral features. Across Layers 1–3 the ictal dynamics are progressively concentrated into a sparse set of high-magnitude, phase-aligned components, whereas interictal responses remain diffuse. The final linear projection separates classes. We also report layer-wise sparsity $S_k = 1 - \|\mathbf{z}\|_1/(\sqrt{n}\,\|\mathbf{z}\|_2)$ and directional phase concentration $C_{\mathrm{dir}} = \left|\frac{1}{|\mathcal{E}|}\sum_{e\in\mathcal{E}} e^{i\phi_e}\right|$. Ictal trials exhibit higher $S_k$ and $C_{\mathrm{dir}}$, consistent with a directed pathological flow enhanced by fractional orders.

## 4 CONCLUSION

EEG-GraphFrFT is a dual-path framework that learns fractional operators in the signal and graph domains to tackle nonstationary EEG. A trainable FrFT adapts its order to track transient, time-varying spectra, while graph fractional operators $e^{-i\theta L^{\alpha}}$ and $e^{-\tau L^{\alpha}}$ model evolving functional connectivity with controlled spectral selectivity. Across three datasets the model shows strong accuracy under colored noise and channel dropouts; ablations verify that learnable orders and the graph path are necessary. Spectral analyses tie seizures to energy concentration on low-order graph modes and increased phase alignment, yielding interpretable biomarkers consistent with clinical hypersynchronization. Hölder stability for fractional graph filters and scalable Chebyshev/Lanczos approximations for $L^{\alpha}$ link the observed robustness to theory with $O(K|E|)$ time and $O(K|V|)$ memory.

Limitations: accuracy relies on wPLI/GC and frequency bands; cross-center and patient metrics await larger tests; graph branch impedes real time.

Outlook: operators generalize to broader complex networks and graph analyses; next steps include self-supervised pretrain, adaptive connectivity, uncertainty calibration, on-device inference.

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

## A  THE USE OF LARGE LANGUAGE MODELS (LLMS)

We used large language models (LLMS) to assist in polishing the language and enhancing the clarity of the manuscript.

## B  APPENDIX

### B.1  MORE RELATED WORK

Recent advances in functional brain network analysis focus on connectivity quantification, graph theory applications, and dynamic network modeling. Construction of functional networks using synchronization indicators like phase lag index (wPLI) (Mao et al., 2022) and coherence analysis (Bowyer, 2016) remains standard, with graph theory providing frameworks for topological analysis (Bullmore and Sporns, 2009). Current research addresses the *threshold problem* in network construction, where threshold selection introduces significant variability in network metrics (Adamovich et al., 2022), prompting data-driven approaches like orthogonal minimum Spanning Tree (OMST) (Dimitriadis et al., 2017). Implicit Neural Representations (INRs) have emerged for modeling dynamic functional connectivity, with (Yu et al., 2017) applying synchronization likelihood in INR frameworks and (Bessadok et al., 2023) developing Graph-Generative Networks (GGNs) for seizure propagation dynamics. Multi-modal integration techniques (Zhang et al., 2021) and challenges in spatial resolution and dynamic modeling persist, as most analyses assume stationarity despite millisecond-scale network reconfigurations being critical for understanding epileptogenesis (Jiang et al., 2023; Bernhardt et al., 2013).

### B.2  MATHEMATICAL FOUNDATIONS

#### B.2.1  WEIGHTED PHASE LAG INDEX (WPLI)

The theoretical motivation for wPLI is that PLI can be unstable under small perturba tions, as small phase fluctuations can flip lag into lead, reducing its sensitivity to weak interactions (Vinck et al., 2011). Similarly, Imaginary Coherency (ImC) normalizes by signal amplitude, making it more sensitive to added uncorrelated noise sources. In contrast, wPLI preserves the numerator of PLI while normalizing by $E[|\Im(S_{ij})|]$, ensuring that $0 \leq \text{wPLI} \leq 1$ and increasing robustness against both noise and volume conduction.

wPLI quantifies non-zero lag phase synchronization between electroencephalogram (EEG) signal channels, effectively suppressing volume conduction effects. For a windowed EEG segment $\mathbf{X} \in \mathbb{R}^{T \times C}$ with $T$ time samples and $C$ channels:

1. Compute the Fourier transforms for each channel at frequency $f$:

$$\mathbf{X}_f = \mathcal{F}(\mathbf{X}) \tag{6}$$

2. Calculate the cross-spectral density between channels $i$ and $j$:

$$S_{ij} = X_i \cdot X_j^* \tag{7}$$

where $*$ denotes complex conjugation.

3. Extract the imaginary part of the cross-spectrum:

$$\Im(S_{ij}) = \textit{imaginary component of } S_{ij} \tag{8}$$

The wPLI between channels $i$ and $j$ at frequency $f$ is defined as:

$$\text{wPLI}_{ij}(f) = \frac{|\mathbb{E}\left[\Im(S_{ij})\right]|}{\mathbb{E}\left[|\Im(S_{ij})|\right]} \tag{9}$$

where $\mathbb{E}$ represents expectation over trials or time windows. This formulation weights phase leads/lags by the magnitude of the cross-spectral imaginary component, emphasizing consistent non-zero lag interactions while suppressing volume-conducted zero-lag synchronization.

### B.2.2 SPECTRAL GRANGER CAUSALITY (sGC)

Spectral GC measures directed frequency-specific information flow between EEG channels (Granger, 1969; Dhamala et al., 2008).

We model the EEG channel pair $x_i(t)$ and $x_j(t)$ with a stable bivariate VAR($p$):

$$\begin{bmatrix} x_i(t) \\ x_j(t) \end{bmatrix} = \sum_{k=1}^{p} A_k \begin{bmatrix} x_i(t-k) \\ x_j(t-k) \end{bmatrix} + \begin{bmatrix} \epsilon_i(t) \\ \epsilon_j(t) \end{bmatrix}, \qquad \Sigma = \text{cov} \begin{bmatrix} \epsilon_i(t) \\ \epsilon_j(t) \end{bmatrix}, \tag{10}$$

where $\mathbf{A}_k = \begin{bmatrix} a_{11}^{(k)} & a_{12}^{(k)} \\ a_{21}^{(k)} & a_{22}^{(k)} \end{bmatrix}$ are coefficient matrices, and $\varepsilon$ are white noise residuals with covariance matrix $\boldsymbol{\Sigma}$.

Define the frequency response

$$H(f) = \left(I - \sum_{k=1}^{p} A_k e^{-i2\pi fk}\right)^{-1} = \begin{bmatrix} H_{ii}(f) & H_{ij}(f) \\ H_{ji}(f) & H_{jj}(f) \end{bmatrix}, \tag{11}$$

so the spectral density is

$$S(f) = H(f)\, \Sigma\, H(f)^* = \begin{bmatrix} S_{ii}(f) & S_{ij}(f) \\ S_{ji}(f) & S_{jj}(f) \end{bmatrix}. \tag{12}$$

To obtain pairwise GC $i \to j$, we orthogonalize the innovations so that $\Sigma$ is diagonal, which yields $\Sigma = \text{diag}(\Sigma_{ii}, \Sigma_{jj})$. Then the total spectrum of $j$ decomposes as

$$S_{jj}(f) = |H_{ji}(f)|^2\, \Sigma_{ii} + |H_{jj}(f)|^2\, \Sigma_{jj}, \tag{13}$$

while the restricted (no-$i$) spectrum is

$$S_{jj|i}(f) = |H_{jj}(f)|^2\, \Sigma_{jj}. \tag{14}$$

Thus the frequency-domain Granger causality reduces to the standard log-ratio:

$$F_{i \to j}(f) = \ln \frac{S_{jj}(f)}{S_{jj|i}(f)} = \ln\left(1 + \frac{|H_{ji}(f)|^2\, \Sigma_{ii}}{|H_{jj}(f)|^2\, \Sigma_{jj}}\right). \tag{15}$$

which matches the widely used formulation of (Dhamala et al., 2008). This is the expression we implement in our pipeline.

For statistical inference, we estimate significance by permutation tests with trial-label shuffling to generate the null distribution of $F_{i \to j}(f)$. To control for multiple comparisons across frequencies, we apply the Benjamini-Hochberg false discovery rate (FDR).

The computation involves three stages:

**Stage 1: Time-domain Vector Autoregressive (VAR) Modeling** For EEG signals $x_i(t)$ and $x_j(t)$, fit a bivariate VAR model:

$$\begin{bmatrix} x_i(t) \\ x_j(t) \end{bmatrix} = \sum_{k=1}^{p} \mathbf{A}_k \begin{bmatrix} x_i(t-k) \\ x_j(t-k) \end{bmatrix} + \begin{bmatrix} \epsilon_i(t) \\ \epsilon_j(t) \end{bmatrix} \tag{16}$$

where $p$ is the model order (determined by AIC/BIC), $\mathbf{A}_k$ are coefficient matrices, and $\epsilon$ are residuals with covariance $\mathbf{\Sigma}$.

**Stage 2: Spectral Transformation** Compute the spectral transfer matrix:

$$\mathbf{H}(f) = \left( \mathbf{I} - \sum_{k=1}^{p} \mathbf{A}_k e^{-i2\pi f k} \right)^{-1} \tag{17}$$

**Stage 3: Frequency-domain GC Calculation** The spectral GC from $i$ to $j$ at frequency $f$ is:

$$\text{sGC}_{i \to j}(f) = \ln \left( 1 + \frac{|H_{ji}(f)|^2 \Sigma_{ii}}{|H_{jj}(f)|^2 \Sigma_{jj}} \right) \tag{18}$$

where $\mathbf{\Sigma} = \text{cov}(\epsilon)$ is the residual covariance matrix.

The final GC metric integrates directional influence across frequency band $F$:

$$\text{GC}_{i \to j} = \frac{1}{|F|} \sum_{f \in F} \text{sGC}_{i \to j}(f) \tag{19}$$

Higher values indicate stronger directed causal influence from channel $i$ to $j$.

### B.2.3 CONNECTIVITY GRAPH CONSTRUCTION

Dynamic functional connectivity networks are constructed using a sliding window approach with a window size of $W = 1024$ samples (2 seconds at 512 Hz) and stride $S = 512$ samples (50% overlap). For each window, the process involves computing the wPLI matrix $\mathbf{W} \in \mathbb{R}^{C \times C}$ to capture undirected phase synchronization, followed by the GC tensor $\mathbf{G} \in \mathbb{R}^{F \times C \times C}$ for $F$ frequency bands (delta: 1–4 Hz, theta: 4–8 Hz, alpha: 8–13 Hz, beta: 13–25 Hz, gamma: 26–80 Hz). Discriminative connections are enhanced through element-wise exponential modulation: $\mathbf{W}_{ij} \leftarrow \mathbf{W}_{ij} \cdot \exp\left(-\gamma_w \cdot \mathbb{I}(\mathbf{W}_{ij} < \tau_w)\right)$ and $\mathbf{G}_{fij} \leftarrow \mathbf{G}_{fij} \cdot \exp\left(-\gamma_g \cdot \mathbb{I}(\mathbf{G}_{fij} < \tau_g)\right)$, where $\gamma_w, \gamma_g$ are learnable decay parameters and $\tau_w, \tau_g$ are adaptive thresholds. The resulting network representations, capturing spatiotemporal dynamics of epileptic propagation, are given by $\mathcal{G} = \left\{ \mathbf{W}^{(t)}, \mathbf{G}^{(t)} \right\}_{t=1}^{N}$, where $N$ is the number of windows, $\mathbf{W}^{(t)}$ is the wPLI network at window $t$, and $\mathbf{G}^{(t)}$ is the multi-band GC tensor.

### B.2.4 FAST FRFT COMPUTATION

The standard Ozaktas-style (Ozaktas et al., 1996) factorization for the fast FRFT:

$$\texttt{frft\_ozaktas}(x, \alpha) = \text{scale} \cdot \text{post\_chirp} \cdot \mathcal{F}^{-1} \left[ \mathcal{F}[\text{pre\_chirp} \cdot x] \odot \mathcal{F}[g] \right] \cdot \text{post\_chirp} \tag{20}$$

The pre-chirp and post-chirp terms are defined as:

$$\text{pre\_chirp}(t) = \exp(-j\pi tan\frac{\phi}{2} \cdot t^2), \quad \text{post\_chirp}(u) = \exp(-j\pi tan\frac{\phi}{2} \cdot u^2) \tag{21}$$

Here, $\phi = \alpha(\pi/2)$. The convolution kernel is

$$h[k] = \frac{1}{\sqrt{|\sin \varphi|}} \exp\left( j\pi \frac{k^2}{\sin \varphi} \right),$$

implemented as a linear convolution via FFT with appropriate zero-padding.

Special cases are handled explicitly, without invoking the general FRFT kernel:

$$\alpha = 0: \quad \text{identity transform} \tag{22}$$
$$\alpha = 1: \quad \text{standard Fourier transform} \tag{23}$$
$$\alpha = -1: \quad \text{inverse Fourier transform} \tag{24}$$
$$\alpha = 2: \quad \text{time reversal} \tag{25}$$

The use of $\tan(\varphi/2)$ in the chirp factors avoids divergence near $\sin\varphi \to 0$, where $\cot\varphi$ is unstable. The kernel scaling $1/\sqrt{|\sin\varphi|}$ ensures unitarity up to a global phase. Near singular angles ($\varphi \approx 0, \pm\pi/2, \pi$), the algorithm snaps to the corresponding explicit branches.

### B.2.5 THEORETICAL PROPERTIES OF GFRFT

**Theorem 3** (Unitarity Preservation). When $F_G$ is unitary ($F_G^H F_G = I$), its eigenvalues satisfy $|\lambda_i| = 1$. The fractional power becomes:

$$\lambda_i^a = e^{ja\theta_i}, \quad \lambda_i = e^{j\theta_i} \tag{26}$$

Thus:

$$(F_G^a)^H F_G^a = V(\Lambda^a)^H \Lambda^a V^{-1} = V\text{diag}(|\lambda_i^a|^2)V^{-1} = I. \tag{27}$$

For the undirected path, the construction is based on the symmetric Laplacian $L_{sym} = U\Lambda U^T$, where $U$ is orthogonal and $\Lambda \geq 0$. The corresponding fractional operator is then defined spectrally as

$$L_{sym}^\alpha = U\Lambda^\alpha U^T, \tag{28}$$

which is guaranteed to be symmetric positive semidefinite but not necessarily unitary.

We have added the difference by distinguishing:

- The unitary case, where $F_G$ itself is unitary and fractional powers preserve unitarity;
- The Laplacian-based case, where $L_{sym}$ or $L_H$ is diagonalized and fractional powers preserve Hermiticity and positive semidefiniteness, but not unitarity.

**Theorem 4** (Index Additivity). Let $F_G^a = \exp(aM)$ and $F_G^b = \exp(bM)$ where $M = \log(F_G)$. Since $M$ commutes with itself:

$$F_G^a F_G^b = \exp(aM)\exp(bM) = \exp((a+b)M) = F_G^{a+b} \tag{29}$$

### B.2.6 EXTENDED FRACTIONAL CALCULUS ON GRAPHS

**Definition 2** (Fractional Graph Derivative). The fractional derivative operator of order $\beta$ is defined as:

$$\mathcal{D}^\beta = F_G^{-\beta}\Lambda_\beta F_G^\beta, \qquad \Lambda_\beta = \text{diag}((\lambda_k)^\beta), \tag{30}$$

where $\lambda_k$ are graph frequencies. This generalizes classical fractional calculus to irregular domains.

**Proposition 1.** Let $W_s$ be the symmetrized adjacency matrix, and let $\Gamma_q = (\Upsilon_q(i,j))$ be a modulation matrix satisfying the conjugate-symmetry condition

$$\Upsilon_q(i,j) = \overline{\Upsilon_q(j,i)} \quad \forall i,j. \tag{31}$$

Define the directed Hermitian Laplacian

$$L_H = D_s - \Gamma_q \odot W_s, \qquad D_s(i,i) = \sum_j W_s(i,j). \tag{32}$$

Then:

1. $L_H$ is Hermitian, i.e. $L_H = L_H^*$.

2. $L_H$ is positive semidefinite, i.e. $x^* L_H x \geq 0$ for all $x \in \mathbb{C}^n$.

3. In the undirected case ($w_{ij} = w_{ji}$, $\Gamma_q \equiv 1$), one recovers $L_H = L_{\text{sym}}$, the standard symmetric Laplacian.

4. By Gershgorin's theorem, all eigenvalues of $L_H$ lie on the nonnegative real axis; hence the spectrum is bounded below by zero.

*Proof.* Since $W_s$ is symmetric with nonnegative entries and $\Upsilon_q(i,j) = \overline{\Upsilon_q(j,i)}$, the off-diagonal entries satisfy

$$L_H(i,j) = -\Gamma_q(i,j)\,W_s(i,j), \qquad L_H(j,i) = -\overline{\Gamma_q(i,j)}\,W_s(i,j), \tag{33}$$

so $L_H(j,i) = \overline{L_H(i,j)}$. Diagonal entries are real, thus $L_H = L_H^*$ (Yan and li, 2023; Wei and Yun, 2024).

For any $x \in \mathbb{C}^n$,

$$x^* L_H x = x^* D_s x - x^*(\Gamma_q \odot W_s)x = \sum_i \Big( \sum_j W_s(i,j) \Big)|x(i)|^2 - \sum_{i,j} W_s(i,j)\,\Upsilon_q(i,j)\,\overline{x(i)}\,x(j). \tag{34}$$

Using $W_s(i,j) = W_s(j,i)$ and $\Upsilon_q(j,i) = \overline{\Upsilon_q(i,j)}$, group $(i,j)$ and $(j,i)$:

$$x^* L_H x = \tfrac{1}{2}\sum_{i,j} W_s(i,j)\Big(|x(i)|^2 + |x(j)|^2 - \Upsilon_q(i,j)\,\overline{x(i)}x(j) - \Upsilon_q(j,i)\,\overline{x(j)}x(i)\Big)$$

$$= \tfrac{1}{2}\sum_{i,j} W_s(i,j)\Big(|x(i)|^2 + |x(j)|^2 - 2\,\Re\{\Upsilon_q(i,j)\,\overline{x(i)}x(j)\}\Big) \tag{35}$$

$$= \tfrac{1}{2}\sum_{i,j} W_s(i,j)\,\big|\,x(i) - \Upsilon_q(i,j)\,x(j)\,\big|^2 \;\geq 0.$$

Hence $L_H \succeq 0$; see also Proposition 2 in (Yan and li, 2023). In the undirected case ($\Gamma_q \equiv 1$), equation 35 reduces to the standard proof for $L_{\text{sym}}$ (Chung, 2005). Finally, since $L_H$ is Hermitian, Gershgorin's theorem applies and all eigenvalues are real and nonnegative.

By Gershgorin's circle theorem, each eigenvalue $\lambda$ of $L_H$ lies in a disk centered at $L_H(i,i) \geq 0$ with radius $\sum_{j \neq i} |L_H(i,j)| = L_H(i,i)$. Thus $\Re(\lambda) \geq 0$, and since $L_H$ is Hermitian, $\lambda \geq 0$. $\square$

For any diagonalization $L_H = UVU^*$ with eigenvalue matrix $V = \text{diag}(\nu_k)$, we define

$$\log V = \text{diag}\big(\log|\nu_k| + j\arg(\nu_k)\big), \quad \nu_k^\alpha = \exp\big(\alpha \log \nu_k\big), \tag{36}$$

using the principal branch $\arg(\nu_k) \in (-\pi, \pi]$. Zero eigenvalues are shifted by a small $\varepsilon$ before applying log or powers, and tiny negative parts due to numerical roundoff are clamped to zero. Thus $\log V$ and $V^\alpha$ are well defined and numerically stable.

### B.2.7 Stability of Graph Fractional Filters under Structural Perturbations

**Setting and notation.** Let $L \in \mathbb{R}^{n \times n}$ be symmetric positive semidefinite (PSD) with spectral decomposition $L = U\,\Lambda\,U^\top$, $\Lambda = \text{diag}(\lambda_1, \ldots, \lambda_n)$ and $0 \leq \lambda_i \leq \Lambda_{\max}$. For $\alpha \in (0, 1]$ and $\tau > 0$ we consider the *fractional heat kernel* (graph fractional filter)

$$\mathcal{H}_{\tau,\alpha}(L) = U\,e^{-\tau \Lambda^\alpha}\,U^\top = e^{-\tau L^{alpha}}. \tag{37}$$

For a perturbed Laplacian $\tilde{L} = L + \Delta \succeq 0$ assume $\|\Delta\|_2 \leq \varepsilon$. Throughout, $\|\cdot\|_2$ denotes the spectral (operator) norm. For a Borel set $I \subseteq [0, \infty)$ we write $P_I(L)$ for the spectral projector of $L$ onto eigenvalues in $I$.

**Preliminaries**

We collect two standard facts used repeatedly.

**Lemma 1** (Operator Hölder continuity of fractional powers). Let $A, B \succeq 0$ and $\alpha \in (0, 1)$. There exists a constant $C_\alpha$ depending only on $\alpha$ (and, if desired, on a priori bounds for $\|A\|_2$ and $\|B\|_2$) such that

$$\|A^\alpha - B^\alpha\|_2 \;\leq\; C_\alpha\,\|A - B\|_2^\alpha. \tag{38}$$

*Proof.* This is classical; see, e.g., the (Birman and Solomyak, 1967) inequalities via double operator integrals or integral representations of fractional powers. Standard references include (Bhatia, 1997) and (Higham, 2008). $\square$

**Lemma 2** (Difference of exponentials). *If $X, Y \succeq 0$ are Hermitian PSD and $\tau > 0$, then*

$$\left\| e^{-\tau X} - e^{-\tau Y} \right\|_2 \leq \tau \left\| X - Y \right\|_2. \tag{39}$$

*Proof.* Consider $F(\theta) = e^{-\tau(1-\theta)X} e^{-\tau\theta Y}$ for $\theta \in [0, 1]$. Then

$$\frac{d}{d\theta} F(\theta) = \tau e^{-\tau(1-\theta)X} (X - Y) e^{-\tau\theta Y}. \tag{40}$$

Hence

$$e^{-\tau X} - e^{-\tau Y} = -\tau \int_0^1 e^{-\tau(1-\theta)X} (X - Y) e^{-\tau\theta Y} \, d\theta. \tag{41}$$

Taking norms and using $\left\| e^{-\tau(1-\theta)X} \right\|_2 \leq 1$ and $\left\| e^{-\tau\theta Y} \right\|_2 \leq 1$ for PSD $X, Y$ yields equation 39. $\square$

**Main stability bound**

**Theorem 5** (Hölder stability of fractional graph filters). *Let $L, \tilde{L} \succeq 0$, $\alpha \in (0, 1]$, and $\tau > 0$. Then*

$$\left\| \mathcal{H}_{\tau,\alpha}(L) - \mathcal{H}_{\tau,\alpha}(\tilde{L}) \right\|_2 \leq \tau \, C_\alpha \left\| L - \tilde{L} \right\|_2^{\min\{1,\alpha\}}. \tag{42}$$

*In particular, for $\alpha \in (0, 1)$ we have the Hölder-$\alpha$ bound $\left\| e^{-\tau L^\alpha} - e^{-\tau \tilde{L}^\alpha} \right\|_2 \leq \tau C_\alpha \left\| L - \tilde{L} \right\|_2^\alpha$, while for $\alpha = 1$ we recover the Lipschitz estimate $\left\| e^{-\tau L} - e^{-\tau \tilde{L}} \right\|_2 \leq \tau \left\| L - \tilde{L} \right\|_2$.*

*Proof.* For $\alpha = 1$, apply Lemma 2 with $X = L, Y = \tilde{L}$. For $\alpha \in (0, 1)$,

$$\left\| e^{-\tau L^\alpha} - e^{-\tau \tilde{L}^\alpha} \right\|_2 \leq \tau \left\| L^\alpha - \tilde{L}^\alpha \right\|_2 \quad \text{by Lemma 2,} \tag{43}$$

then use Lemma 1 to obtain

$$\left\| L^\alpha - \tilde{L}^\alpha \right\|_2 \leq C_\alpha \left\| L - \tilde{L} \right\|_2^\alpha. \tag{44}$$

$\square$

**Remark 1** (Directed graphs via Hermitianization). If a directed functional network is represented by a Hermitian PSD operator $L_H$ (e.g., via a conjugate-symmetric modulation of a symmetrized adjacency), then the same analysis applies verbatim to $L_H$ and its perturbations $\tilde{L}_H$, since the functional calculus and the contraction property of $e^{-\tau X}$ require only $X \succeq 0$.

**High-frequency attenuation on spectral subspaces**

We make the high-frequency attenuation precise on a spectral subspace $E$ of $L$.

**Lemma 3** (Quadratic-form lower bound on $E$). Let $\tilde{L} = L + \Delta \succeq 0$ with $\|\Delta\|_2 \leq \varepsilon$, and fix $\lambda_0 > 0$. Let $P_E := P_{[\lambda_0, \infty)}(L)$ be the spectral projector of $L$ onto eigenvalues $\geq \lambda_0$. Then for any $v \in E = \operatorname{Ran}(P_E)$ and $\alpha \in (0, 1]$,

$$v^\top \tilde{L} v \geq v^\top L v - \|\Delta\|_2 \|v\|_2^2 \geq (\lambda_0 - \varepsilon) \|v\|_2^2, \tag{45}$$

$$v^\top \tilde{L}^\alpha v \geq (\lambda_0 - \varepsilon)_+^\alpha \|v\|_2^2, \tag{46}$$

whence, for any $s \in [0, 1]$ and $\tau > 0$,

$$\left\| e^{-\tau s \tilde{L}^\alpha} P_E \right\|_2 \leq e^{-\tau s (\lambda_0 - \varepsilon)_+^\alpha}, \qquad \left\| P_E e^{-\tau(1-s)L^\alpha} \right\|_2 \leq e^{-\tau(1-s)\lambda_0^\alpha}. \tag{47}$$

*Proof.* Inequality equation 45 follows from $v^\top \Delta v \geq -\|\Delta\|_2 \|v\|_2^2$ and $v^\top L v \geq \lambda_0 \|v\|_2^2$ on $E$. Operator monotonicity of $t \mapsto t^\alpha$ on $[0, \infty)$ gives equation 46. The bounds in equation 47 then follow by the spectral theorem, since $e^{-\tau s \tilde{L}^\alpha} \preceq e^{-\tau s(\lambda_0 - \varepsilon)_+^\alpha} I$ on $E$ while $L^\alpha \succeq \lambda_0^\alpha$ on $E$. $\square$

**Theorem 6** (Exponential attenuation on high-frequency subspaces). *Let $L, \tilde{L} \succeq 0$ with $\tilde{L} = L + \Delta$, $\|\Delta\|_2 \leq \varepsilon$, and let $P_E = P_{[\lambda_0, \infty)}(L)$ with $\lambda_0 > 0$. For $\alpha \in (0, 1]$ and $\tau > 0$,*

$$\left\| P_E \big( e^{-\tau L^\alpha} - e^{-\tau \tilde{L}^\alpha} \big) P_E \right\|_2 \leq \tau \, e^{-\tau(\lambda_0 - \varepsilon)_+^\alpha} \left\| L^\alpha - \tilde{L}^\alpha \right\|_2. \tag{48}$$

Consequently, by Lemma 1,

$$\left\| P_E \big( e^{-\tau L^\alpha} - e^{-\tau \tilde{L}^\alpha} \big) P_E \right\|_2 \leq \tau \, C_\alpha \, e^{-\tau(\lambda_0 - \varepsilon)_+^\alpha} \left\| L - \tilde{L} \right\|_2^{\min\{1, \alpha\}}. \tag{49}$$

*Proof.* Apply the integral identity of Lemma 2 with $X = L^\alpha$ and $Y = \tilde{L}^\alpha$ and insert projectors:

$$P_E \big( e^{-\tau L^\alpha} - e^{-\tau \tilde{L}^\alpha} \big) P_E = -\tau \int_0^1 P_E e^{-\tau(1-s)L^\alpha} (L^\alpha - \tilde{L}^\alpha) e^{-\tau s \tilde{L}^\alpha} P_E \, ds. \tag{50}$$

Taking norms and using equation 47 gives

$$\left\| P_E (e^{-\tau L^\alpha} - e^{-\tau \tilde{L}^\alpha}) P_E \right\|_2 \leq \tau \left\| L^\alpha - \tilde{L}^\alpha \right\|_2 \int_0^1 e^{-\tau(1-s)\lambda_0^\alpha} e^{-\tau s(\lambda_0 - \varepsilon)_+^\alpha} \, ds \tag{51}$$

$$\leq \tau \left\| L^\alpha - \tilde{L}^\alpha \right\|_2 e^{-\tau(\lambda_0 - \varepsilon)_+^\alpha}, \tag{52}$$

where the final step bounds the integral by the maximum of the integrand (attained at $s = 1$ since $(\lambda_0 - \varepsilon)_+^\alpha \leq \lambda_0^\alpha$). $\qquad\square$

**Remark 2** (Perturbations confined to $E$). If $\Delta = P_E \Delta P_E$ (no cross-coupling between $E$ and $E^\perp$), then $E$ reduces both $L$ and $\tilde{L}$, and

$$\left\| P_E \big( e^{-\tau L^\alpha} - e^{-\tau \tilde{L}^\alpha} \big) P_E \right\|_2 = \left\| e^{-\tau(L|_E)^\alpha} - e^{-\tau(L|_E + \Delta|_E)^\alpha} \right\|_2 \leq \tau \, C_\alpha \, \|P_E \Delta P_E\|_2^{\min\{1, \alpha\}} \tag{53}$$

by combining Lemmas 2 and 1. For general $\Delta$, Theorem 6 shows an additional *exponential* attenuation $e^{-\tau(\lambda_0 - \varepsilon)_+^\alpha}$ on $E$.

**Corollary 1** (Global-to-local high-frequency control). With the assumptions of Theorem 6,

$$\left\| \mathcal{H}_{\tau, \alpha}(L) - \mathcal{H}_{\tau, \alpha}(\tilde{L}) \right\|_2 \geq \left\| P_E \big( \mathcal{H}_{\tau, \alpha}(L) - \mathcal{H}_{\tau, \alpha}(\tilde{L}) \big) P_E \right\|_2 \leq \tau \, C_\alpha \, e^{-\tau(\lambda_0 - \varepsilon)_+^\alpha} \left\| L - \tilde{L} \right\|_2^{\min\{1, \alpha\}}. \tag{54}$$

Thus perturbations that act predominantly on high-eigenvalue subspaces are strongly suppressed by the factor $e^{-\tau(\lambda_0 - \varepsilon)_+^\alpha}$.

**Implications: Robustness of graph FrFT and derived operators**

We explain how the preceding results provide precise robustness guarantees for the graph fractional Fourier transform (FrFT) $\mathcal{F}_{\theta, \alpha}(L) = e^{-i\theta L^\alpha}$ and the unified complex-time family $\mathcal{G}_{\tau, \theta, \alpha}(L) = e^{-(\tau + i\theta)L^\alpha}$, $\tau \geq 0$.

**Proposition 2** (Structural robustness of FrFT). For $L, \tilde{L} \succeq 0$, $\alpha \in (0, 1]$, $\theta \in \mathbb{R}$,

$$\|\mathcal{F}_{\theta, \alpha}(L) - \mathcal{F}_{\theta, \alpha}(\tilde{L})\|_2 \leq |\theta| \, C_\alpha \, \|L - \tilde{L}\|_2^{\min\{1, \alpha\}}. \tag{55}$$

**Corollary 2** (Complex-time family: stability and high-frequency damping). For $\mathcal{G}_{\tau, \theta, \alpha}$,

$$\|\mathcal{G}_{\tau, \theta, \alpha}(L) - \mathcal{G}_{\tau, \theta, \alpha}(\tilde{L})\|_2 \leq \sqrt{\tau^2 + \theta^2} \, C_\alpha \, \|L - \tilde{L}\|_2^{\min\{1, \alpha\}}. \tag{56}$$

If $\tau > 0$ and $P_E = P_{[\lambda_0, \infty)}(L)$, then

$$\|P_E(\mathcal{G}_{\tau, \theta, \alpha}(L) - \mathcal{G}_{\tau, \theta, \alpha}(\tilde{L}))P_E\|_2 \leq \sqrt{\tau^2 + \theta^2} \, C_\alpha \, e^{-\tau(\lambda_0 - \varepsilon)_+^\alpha} \, \|L - \tilde{L}\|_2^{\min\{1, \alpha\}}. \tag{57}$$

**Derived operators.** (i) Many graph time–frequency atoms have the form $g(L) = e^{-(\tau + i\theta)L^\alpha}$ and inherit the bounds above. (ii) For the fractional resolvent $\mathcal{R}_{\tau, \alpha}(L) = (I + \tau L^\alpha)^{-1}$,

$$\|\mathcal{R}_{\tau, \alpha}(L) - \mathcal{R}_{\tau, \alpha}(\tilde{L})\|_2 \leq \tau \, C_\alpha \, \|L - \tilde{L}\|_2^{\min\{1, \alpha\}}. \tag{58}$$

(iii) Input-noise robustness: $\|\mathcal{F}_{\theta, \alpha}(L)\|_2 = 1$ and $\|\mathcal{G}_{\tau, \theta, \alpha}(L)\|_2 \leq 1$ for $\tau \geq 0$, so these operators do not amplify measurement noise.

### B.2.8 Differentiability of GFRFT

We work specifically with the Hermitian directed Laplacian $L_H = D_s - \Gamma_q \odot W_s$, under the conjugate-symmetry condition $\Upsilon_q(i,j) = \overline{\Upsilon_q(j,i)}$ and $W_s = W_s^\top \geq 0$. Under these assumptions, $L_H = L_H^*$ and $L_H \succeq 0$. [1]

**Definition 3** (Spectral definition of $L_H^\alpha$). Let the eigendecomposition of $L_H$ be

$$L_H = U \Lambda U^H, \qquad U^H U = I, \quad \Lambda = \mathrm{diag}(\nu_1, \ldots, \nu_n), \ \nu_k \geq 0. \tag{59}$$

For any $\alpha \in \mathbb{R}$, the fractional power of $L_H$ is defined by spectral calculus:

$$L_H^\alpha = U \Lambda^\alpha U^H, \qquad \Lambda^\alpha = \mathrm{diag}(\nu_1^\alpha, \ldots, \nu_n^\alpha), \tag{60}$$

with the convention $0^\alpha = 0$ for $\alpha > 0$.

**Well-posedness.** Since $L_H$ is Hermitian PSD, equation 60 is well-defined for all $\alpha \geq 0$. When derivatives with respect to $\alpha$ are required and zero eigenvalues are present, we use the standard $\varepsilon$-shift $\Lambda_\varepsilon = \Lambda + \varepsilon I$ and set $L_{H,\varepsilon}^\alpha = U \Lambda_\varepsilon^\alpha U^H$, then take $\varepsilon \downarrow 0$ after differentiation. This matches the principal-branch matrix-function calculus used in the paper for logarithm and exponentiation.

**Proposition 3** (Basic properties). Let $L_H = U\Lambda U^H$ and $L_H^\alpha$ be defined by equation 60. For $\alpha, \beta \geq 0$:

1. **PSD and Hermitianity:** $L_H^\alpha = (L_H^\alpha)^H \succeq 0$.

2. **Semigroup (index additivity):** $L_H^\alpha L_H^\beta = L_H^{\alpha+\beta}$.

3. **Spectral monotonicity:** if $0 \leq \alpha \leq \beta$ then $L_H^\alpha \preceq L_H^\beta$.

4. **Continuity in $\alpha$:** $\alpha \mapsto L_{H,\varepsilon}^\alpha$ is continuous and differentiable for every $\varepsilon > 0$, with derivative

$$\frac{\partial}{\partial \alpha} L_{H,\varepsilon}^\alpha = U \left( \Lambda_\varepsilon^\alpha \odot \log \Lambda_\varepsilon \right) U^H, \tag{61}$$

where "$\odot$" denotes the Hadamard product and $\log \Lambda_\varepsilon = \mathrm{diag}(\log(\nu_k + \varepsilon))$.

*Proof.* All items follow from spectral calculus on the real nonnegative spectrum.

(1) Hermitianity is immediate since $U\Lambda^\alpha U^H$ with $\Lambda^\alpha$ real diagonal is Hermitian; PSD holds because each eigenvalue $\nu_k^\alpha \geq 0$.

(2) Using equation 60, $L_H^\alpha L_H^\beta = U \Lambda^\alpha U^H U \Lambda^\beta U^H = U \Lambda^{\alpha+\beta} U^H = L_H^{\alpha+\beta}$.

(3) For $0 \leq \alpha \leq \beta$, the scalar function $t \mapsto t^\gamma$ on $t \geq 0$ is monotone increasing in $\gamma$. Hence $\nu_k^\alpha \leq \nu_k^\beta$ for all $k$, which implies $L_H^\alpha \preceq L_H^\beta$.

(4) For $\varepsilon > 0$, each scalar map $\alpha \mapsto (\nu_k + \varepsilon)^\alpha$ is $C^\infty$ with derivative $(\nu_k + \varepsilon)^\alpha \log(\nu_k + \varepsilon)$. Stacking on the diagonal and conjugating by $U$ gives equation 61. $\square$

**Action on features and gradient.** Given features $X \in \mathbb{C}^{n \times d}$, the directed-graph branch applies $L_H^\alpha$ spectrally:

$$Z_{\mathrm{dir}} = L_H^\alpha X = U \Lambda^\alpha U^H X. \tag{62}$$

Let $\mathcal{L}$ be any real-valued loss on $Z_{\mathrm{dir}}$. Using equation 61 with an $\varepsilon$-shift, the gradient w.r.t. $\alpha$ is

$$\frac{\partial \mathcal{L}}{\partial \alpha} = \mathrm{Re}\,\mathrm{Tr}\left( \left( \frac{\partial \mathcal{L}}{\partial Z_{\mathrm{dir}}} \right)^H U \left( \Lambda_\varepsilon^\alpha \circ \log \Lambda_\varepsilon \right) U^H X \right), \tag{63}$$

where the real part ensures a real gradient for complex-valued intermediates. This matches the matrix-function identities already used in the manuscript for $F_G^a = \exp(a \log F_G)$ and $\log(F_G^a) = a \log(F_G)$ under principal-branch assumptions.

---

[1] Hermitian and PSD properties follow from the quadratic form $\frac{1}{2} \sum_{i,j} W_s(i,j) |x_i - \Upsilon_q(i,j) x_j|^2 \geq 0$.

### B.2.9 BILINEAR FUSION DETAILS

The general bilinear interaction between feature vectors $\mathbf{s} \in \mathbb{R}^{d_s}$ (CHAN1) and $\mathbf{g} \in \mathbb{R}^{d_g}$ (CHAN2) is defined as $y_k = \mathbf{s}^\top \mathbf{W}_k \mathbf{g} + b_k$ for $k = 1, \ldots, d_o$, where $\mathbf{W}_k \in \mathbb{R}^{d_s \times d_g}$ is a weight matrix for each output dimension $k$, and $b_k$ is a bias term. The output $\mathbf{y} \in \mathbb{R}^{d_o}$ comprises elements $y_k$. This requires $d_o \cdot d_s \cdot d_g$ parameters, which is costly for high-dimensional features.

To improve efficiency, a low-rank constraint is imposed on $\mathbf{W}_k$, assuming rank $r \ll \min(d_s, d_g)$. Each $\mathbf{W}_k$ is factorized as $\mathbf{W}_k = \mathbf{V}_{s,k} \mathbf{V}_{g,k}^\top$, where $\mathbf{V}_{s,k} \in \mathbb{R}^{d_s \times r}$ and $\mathbf{V}_{g,k} \in \mathbb{R}^{d_g \times r}$. Substituting yields:

$$y_k = \mathbf{s}^\top \mathbf{V}_{s,k} \mathbf{V}_{g,k}^\top \mathbf{g} + b_k = \sum_{i=1}^{r} (\mathbf{V}_{s,k}[i]^\top \mathbf{s})(\mathbf{V}_{g,k}[i]^\top \mathbf{g}) + b_k, \tag{64}$$

or, using the Hadamard product $\odot$:

$$y_k = \mathbf{1}^\top \left[ (\mathbf{V}_{s,k}^\top \mathbf{s}) \odot (\mathbf{V}_{g,k}^\top \mathbf{g}) \right] + b_k. \tag{65}$$

To further reduce parameters, shared projection matrices $\mathbf{V}_s \in \mathbb{R}^{d_s \times r}$ and $\mathbf{V}_g \in \mathbb{R}^{d_g \times r}$ are used, with an output projection matrix $\mathbf{U} \in \mathbb{R}^{r \times d_o}$. The final output is:

$$\mathbf{y} = \mathbf{U}^\top \left[ (\mathbf{V}_s^\top \mathbf{s}) \odot (\mathbf{V}_g^\top \mathbf{g}) \right] + \mathbf{b}. \tag{66}$$

This reduces parameter complexity from $\mathcal{O}(d_s \cdot d_g \cdot d_o)$ to $\mathcal{O}((d_s + d_g + d_o) \cdot r)$, ensuring computational efficiency and strong performance.

### B.3 FULL RESULTS

Table A1: Performance comparison of different models on epileptic EEG classification tasks under varying noise conditions for the FMCE dataset. For Pink Noise conditions (SD=0.5-1.0), the **best** and **second-best** percentage changes, indicating noise robustness, are highlighted for each metric.

| Models | Original | | | | | Pink Noise (SD=0.5) | | | | | Pink Noise (SD=1) | | | | |
|---|---|---|---|---|---|---|---|---|---|---|---|---|---|---|---|
| Metrics | Acc | F1 | AUC | Sens | Spec | Acc | F1 | AUC | Sens | Spec | Acc | F1 | AUC | Sens | Spec |
| EEG-Conformer (Song et al., 2023) | 0.8990 | 0.8990 | 0.9578 | 0.8807 | 0.9100 | 0.8792 (-2.2%) | 0.8732 (-2.9%) | 0.8728 (-8.9%) | 0.8667 (-1.6%) | 0.8967 (-1.5%) | 0.8630 (-4.0%) | 0.8636 (-3.9%) | 0.8662 (-9.6%) | 0.8240 (-6.4%) | 0.8910 (-2.1%) |
| Mamba (Gui et al., 2024) | 0.9182 | 0.9183 | 0.9568 | 0.9533 | 0.9246 | 0.8989 (-2.1%) | 0.8890 (-3.2%) | 0.9521 (-0.5%) | 0.9411 (-1.3%) | 0.9100 (-1.6%) | 0.8559 (-6.8%) | 0.8533 (-7.1%) | 0.9337 (-2.4%) | 0.8400 (-11.9%) | 0.8700 (-5.9%) |
| FAPEX (Zheng et al., 2025) | 0.9005 | 0.9003 | 0.9567 | 0.9341 | 0.9588 | 0.9070 (+0.7%) | 0.9073 (+0.8%) | 0.9733 (+1.7%) | 0.8934 (-4.4%) | 0.9487 (-1.1%) | 0.8899 (-1.2%) | 0.8974 (-0.3%) | 0.8976 (-6.2%) | 0.8841 (-5.4%) | 0.9388 (-2.1%) |
| **EEG-GraphFrFT** | **0.9454** | **0.9412** | **0.9756** | **0.9528** | **0.9746** | **0.9404** (-0.5%) | **0.9462** (+0.5%) | **0.9736** (-0.2%) | **0.9451** (-0.8%) | **0.9686** (-0.6%) | **0.9376** (-0.8%) | **0.9393** (-0.2%) | **0.9706** (-0.5%) | **0.9378** (-1.6%) | **0.9696** (-0.5%) |

Table A2: Performance comparison of more different baselines on epileptic EEG classification tasks.

| Models | HUP | | | | | FMCE | | | | | Helsinki Neonatal EEG | | | | |
|---|---|---|---|---|---|---|---|---|---|---|---|---|---|---|---|
| Metrics | Acc | F1 | AUC | Sens | Spec | Acc | F1 | AUC | Sens | Spec | Acc | F1 | AUC | Sens | Spec |
| Scattering Transformer(Zheng et al., 2023) | - | - | - | - | - | - | - | - | - | - | 0.9055 | 0.7990 | 0.9638 | - | - |
| MBGNN(Tang and Zhao, 2024) | - | - | - | - | - | - | - | - | - | - | - | - | 0.9911 | - | - |
| Extra Large (XL) ConvNeXt(Hogan et al., 2025) | - | - | - | - | - | - | - | - | - | - | - | - | 0.9280 | - | - |
| Deep Residual Neural Network(Webb et al., 2021) | - | - | - | - | - | - | - | - | - | - | 0.9550 | 0.9000 | 0.9620 | - | - |
| CNN(Gomez-Quintana et al., 2022) | - | - | - | - | - | - | - | - | - | - | - | - | 0.8640 | 0.8900 | 0.7800 |
| Fd-CAE(Wang et al., 2024) | - | - | - | - | - | - | - | - | - | - | 0.9234 | 0.9577 | - | - | - |
| CNN(Gramacki and Gramacki) | - | - | - | - | - | - | - | - | - | - | 0.9700 | - | - | - | - |
| SRFE(O'Leary et al., 2024) | - | - | - | - | - | 0.9520 | - | 0.9100 | 1.0000 | - | - | - | - | - | - |
| seizure matching system(Thomas et al., 2024) | - | 0.6700 | 0.7600 | - | - | - | - | - | - | - | - | - | - | - | - |
| Random Forest Classifier(Bernabei et al., 2022) | - | - | 0.7700 | - | - | - | - | - | - | - | - | - | - | - | - |
| LASSO Logistic Regression(Conrad et al., 2024) | 0.7930 | - | 0.8300 | - | - | - | - | - | - | - | - | - | - | - | - |
| SDA(Tapani et al., 2019) | - | - | - | - | - | - | - | - | - | - | - | - | 0.9880 | 0.7600 | - |
| 1D FCN,2D FCN(O'Shea et al., 2020) | - | - | - | - | - | - | - | - | - | - | - | - | 0.9000 | - | - |
| Morse Wavelets based LBP coupled with an ensemble classifier(Diykh et al., 2022) | - | - | - | - | - | - | - | - | - | - | 0.9700 | - | 0.9400 | - | - |
| LMA-EEGNet(Zhou et al., 2024) | - | - | - | - | - | - | - | - | - | - | 0.9571 | - | 0.9862 | 0.9500 | - |
| SSGNN + AO(Nelson et al., 2024) | - | - | - | - | - | - | - | - | - | - | 0.9870 | - | 0.9879 | - | - |
| Logistic Regression Model(Conrad et al., 2024) | 0.7930 | - | 0.8300 | - | - | - | - | - | - | - | - | - | - | - | - |
| Simple RF(O'Leary et al., 2024) | - | - | - | - | - | 0.7800 | - | - | 0.1000 | 0.9900 | - | - | - | - | - |
| Averaged RF(O'Leary et al., 2024) | - | - | - | - | - | 0.7900 | - | - | 0.1000 | 0.9900 | - | - | - | - | - |
| Manifold RF(Yu et al., 2021) | - | - | - | - | - | - | - | 0.8900 | - | - | - | - | - | - | - |
| (Scheid et al., 2022) | - | - | 0.8300 | - | - | - | - | - | - | - | - | - | - | - | - |
| Virtual Resection(Bernabei, 2021) | 0.8600 | - | 0.8900 | - | - | - | - | - | - | - | - | - | - | - | - |
| Network-based epilepsy model(Tyner, 2024) | 0.6600 | - | - | 0.8900 | 0.6500 | - | - | - | - | - | - | - | - | - | - |
| functional connectivity networks(Scheid et al., 2021) | - | - | 0.7500 | - | - | - | - | - | - | - | - | - | - | - | - |
| IED-to-LoWS delay–based classifier(Sheybani et al., 2025) | 0.6300 | 0.6700 | 0.6800 | 0.6700 | 0.6000 | - | - | - | - | - | - | - | - | - | - |
| VEP(Shen et al., 2025) | - | - | 0.7500 | - | - | - | - | - | - | - | - | - | - | - | - |
| Neural Fragility model(Millán Vidal et al., 2024) | - | - | - | - | - | 0.7900 | 0.7400 | 0.8200 | - | - | - | - | - | - | - |
| SeizureNet(Janca et al., 2021) | - | - | - | - | - | - | - | - | - | - | 0.8700 | 0.8400 | 0.9000 | - | - |

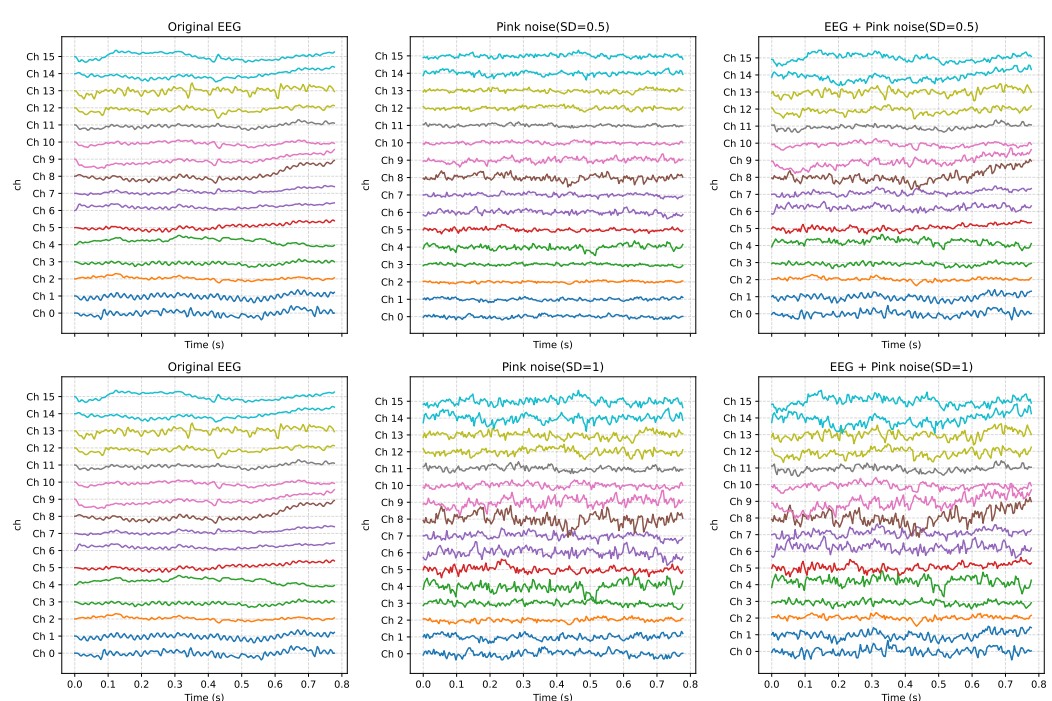

Figure A1: **Visualization comparison of original EEG signals and those superimposed with pink noise at different intensities**. (a) Original clean EEG signals; (b) Pink noise with standard deviation of 0.5; (c) Original EEG signals superimposed with SD=0.5 noise; (d) Pink noise with standard deviation of 1.0; (e) Original EEG signals superimposed with SD=1.0 noise. The figure displays signal waveforms of 16 electrode channels (Ch 0-Ch 15) within the 0-0.8 second time window, with different colors representing different channels. The comparison clearly demonstrates the contamination degree of EEG signals by noise at different intensities, providing visual evidence for evaluating model noise robustness.

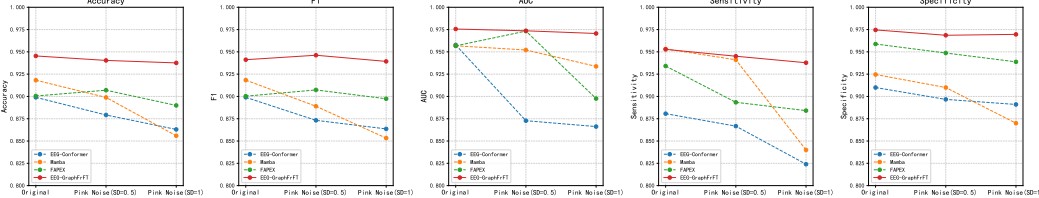

Figure A2: Performance comparison of various models under different noise conditions. The figure displays five metrics (Accuracy, F1 score, AUC, Sensitivity, and Specificity) for EEG-Conformer (blue dashed line), Mamba (orange dashed line), FAPEX (green solid line), and the proposed EEG-GraphFrFT (red solid line) across three scenarios: original EEG (no noise), pink noise with standard deviation 0.5, and pink noise with standard deviation 1.0. The results illustrate the robustness of each model against increasing noise intensity, with EEG-GraphFrFT demonstrating superior stability across all metrics.

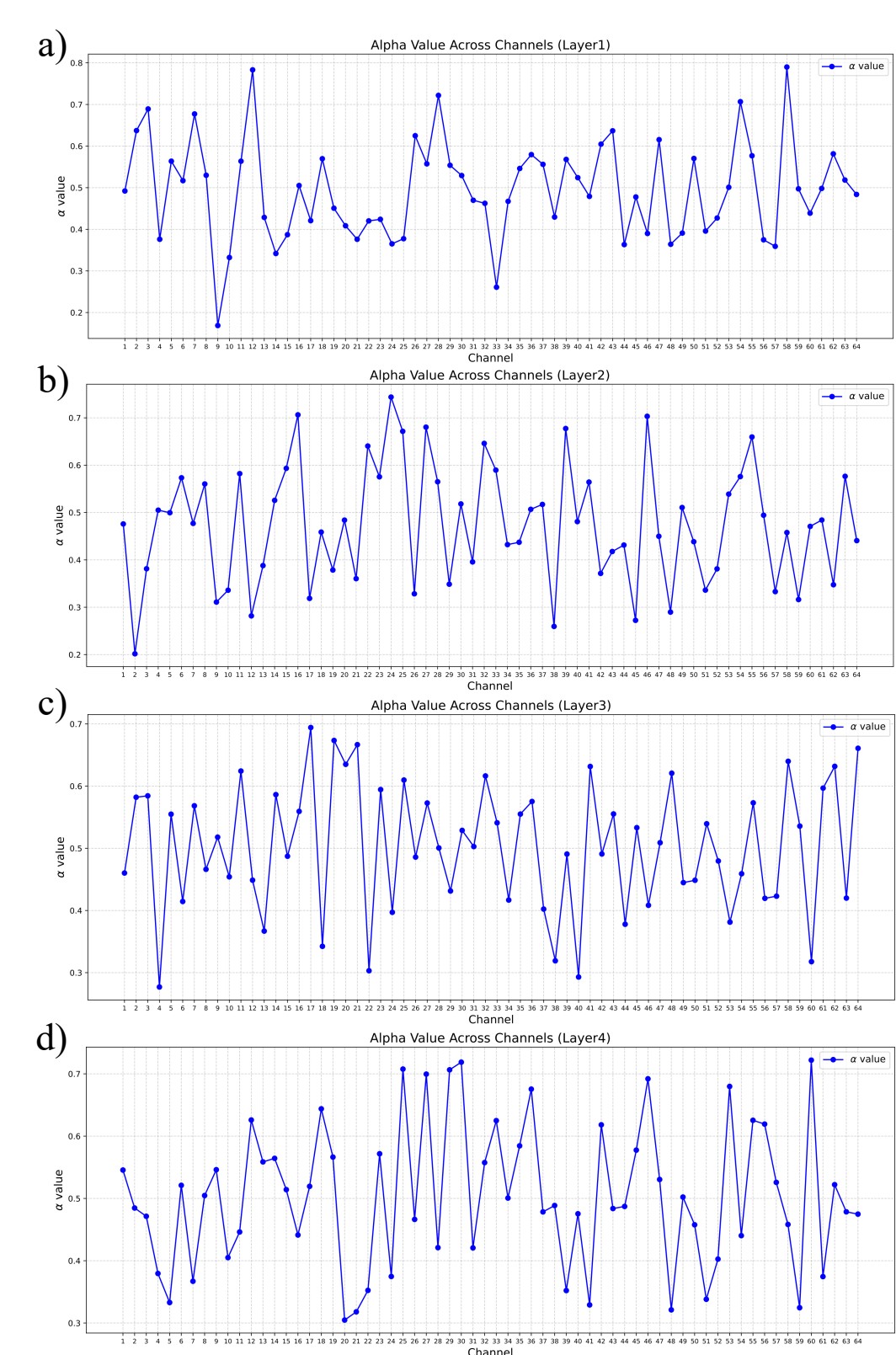

Figure A3: Distribution of learned fractional order parameter $\alpha$ across 64 different channels for four layers (Layer1 to Layer4) in CHAN1. The line plot illustrates the adaptive variation of $\alpha$ values (ranging 0.1-0.8) optimized for each EEG channel, demonstrating the model's capacity to tailor time–frequency resolution to channel-specific characteristics. The non-uniform distribution reflects the adaptive learning of fractional orders for capturing non-stationary patterns in epileptic EEG signals.

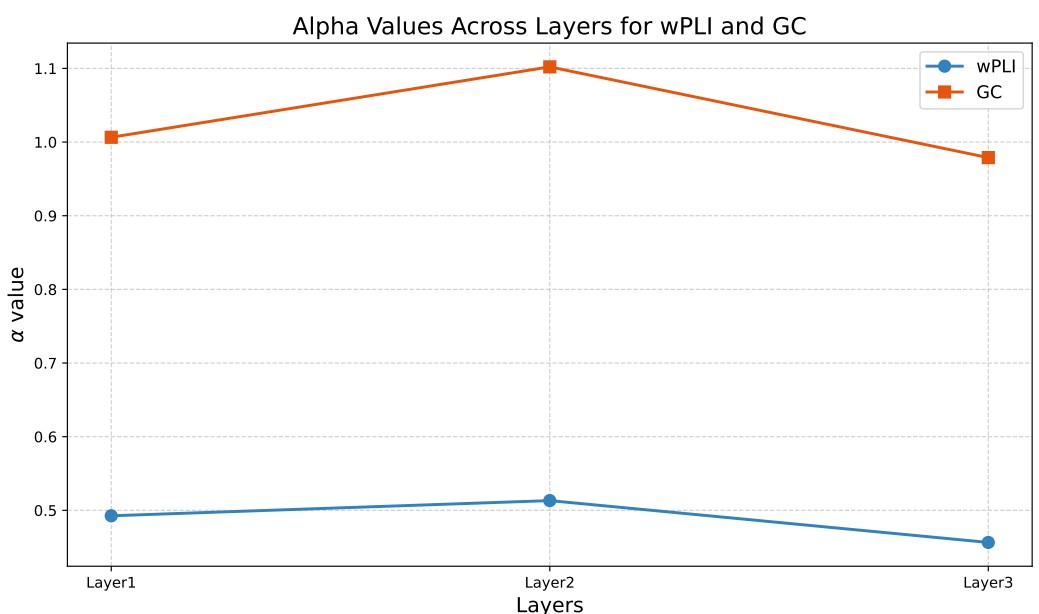

Figure A4: Distribution of the learned fractional order parameter $\alpha$ for the wPLI and GC algorithms across three layers (Layer 1 to Layer 3) in the CHAN2 model. The line plot illustrates the distinct distributions of the $\alpha$ values optimized for the functional networks constructed by the two algorithms. The $\alpha$ values for wPLI remain stable at approximately 0.5 across all layers, whereas the $\alpha$ values for GC are distributed around 1.05. This differential distribution demonstrates the capacity of our trainable GFRFT layer to tailor the time-frequency resolution specifically for the wPLI- and GC-based networks, reflecting the model's adaptive learning for capturing the complex non-stationary dynamics in epileptic EEG signals.

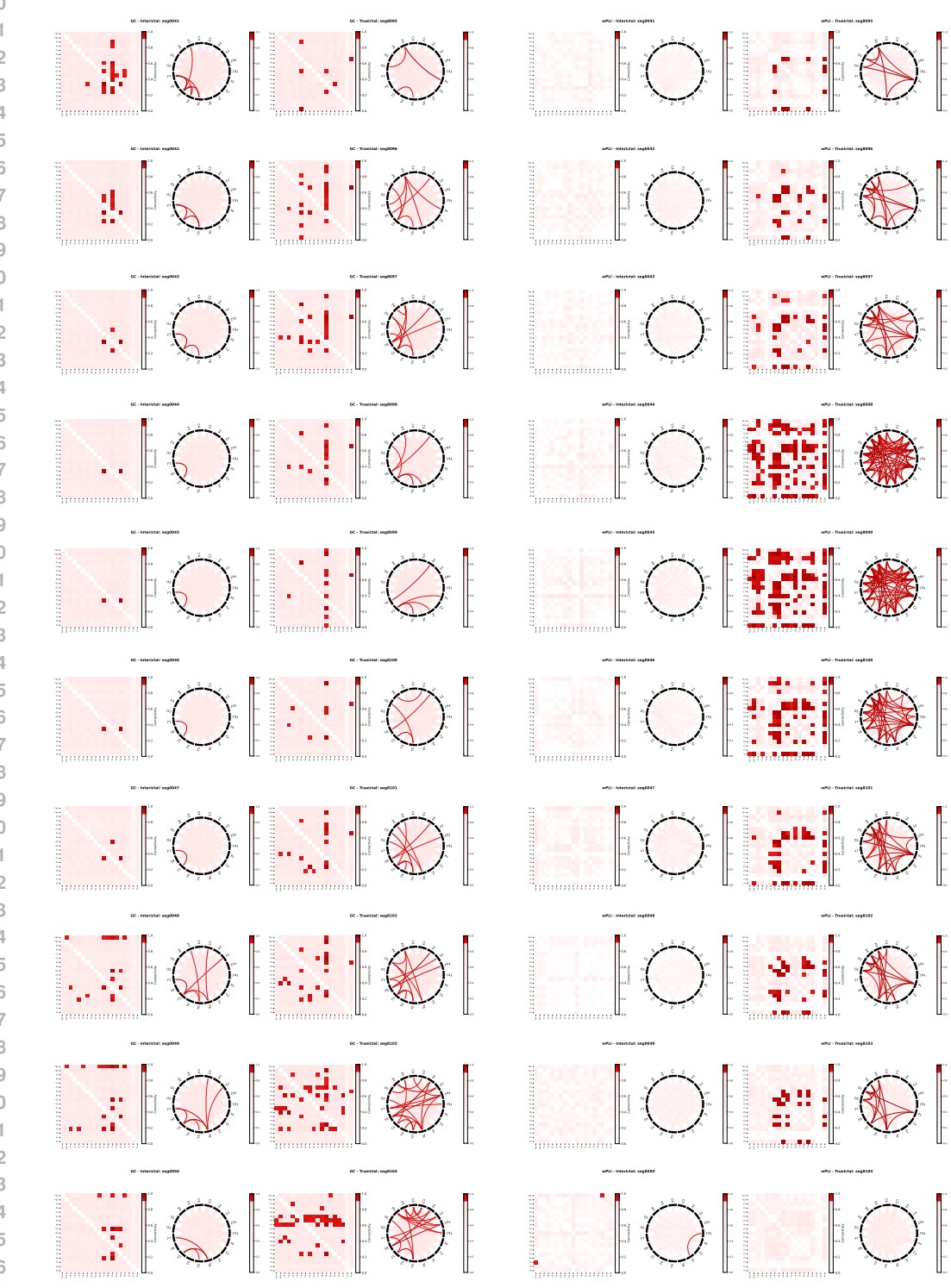

Figure A5: The figure shows functional connectivity matrices and brain network topology diagrams constructed using frequency-domain Granger Causality (GC) for 10 non-seizure samples (left two subfigures) and seizure samples (right two subfigures).

Figure A6: The figure shows functional connectivity matrices and brain network topology diagrams constructed using wPLI for 10 non-seizure samples (left two subfigures) and seizure samples (right two subfigures).

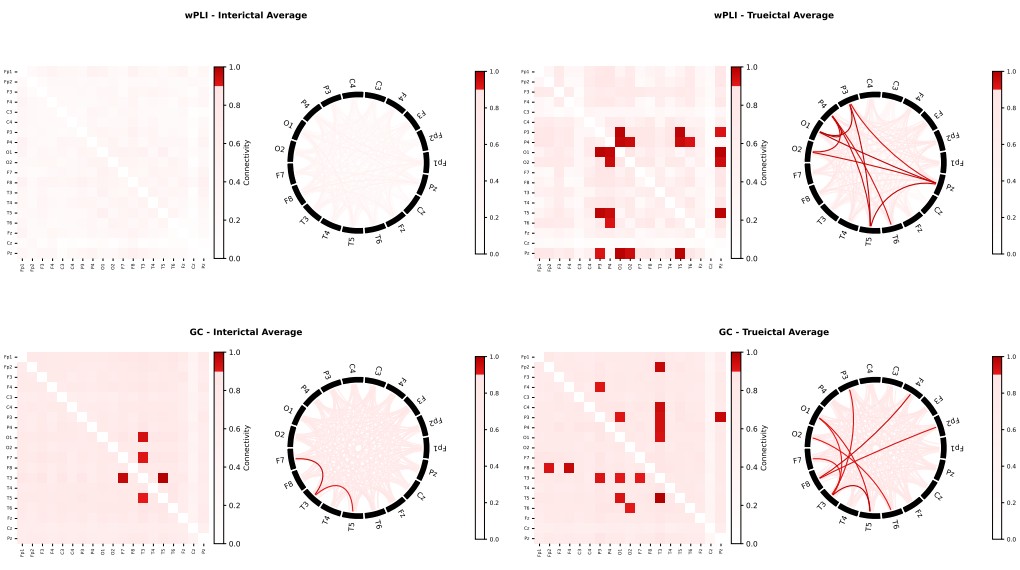

Figure A7: Average Functional connectivity analysis using wPLI and GC. (**Left**) Interictal period: sparse and weak synchronization. (**Right**) Ictal period: significantly enhanced phase-synchronization strength and density, demonstrating epileptic hypersynchronization.

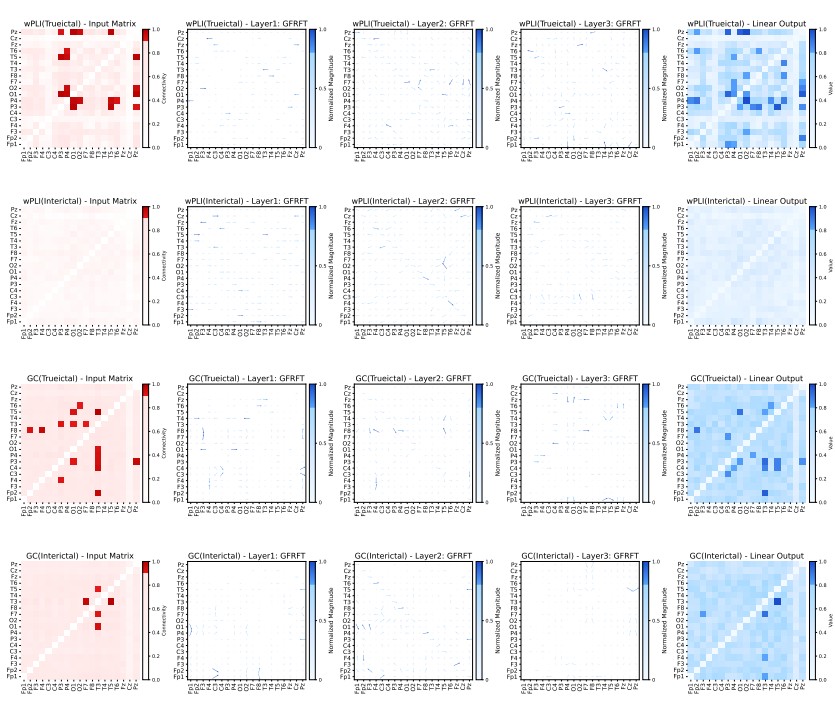

Figure A8: The transformation pipeline of wPLI and frequency-domain Granger Causality (GC) EEG functional connectivity data through the intermediate layers of the Graph Fractional Fourier Transform (GFRFT) is illustrated in the figure.

## REPRODUCIBILITY STATEMENT

This study was conducted under the following computational conditions to ensure reproducibility:

Table A3: Experimental Configuration Details

| Parameter | Value |
|---|---|
| **Hardware** | |
| - GPUs | $2 \times$ NVIDIA A100 (80GB) |
| - CPU | Intel(R) Xeon(R) Platinum 8369B CPU @ 2.90GHz |
| - RAM | 1TB DDR4 |
| | |
| **Software** | |
| - OS | Ubuntu 22.04 LTS |
| - CUDA | 11.8 |
| - cuDNN | 8.9 |
| - Python | 3.10 |
| - PyTorch | 2.1.0 |
| | |
| **Training Configuration** | |
| - Batch size | 16 |
| - Epochs | 100 (with early stopping) |
| - Random seed | 42 |
| - Optimizer | RAdam + Lookahead |
| • betas | (0.9, 0.999) |
| • eps | 1e-8 |
| • lookahead_k | 5 |
| • lookahead_alpha | 0.5 |
| - Learning rate | 5e-4 |
| - LR scheduler | ReduceLROnPlateau |
| • factor | 0.5 |
| • patience | 3 |
| • min_lr | 1e-6 |
| - Weight decay | 1e-4 |
| - Early stopping | |
| • patience | 10 |
| • min_delta | 0.001 |
| - Gradient clipping | Enabled |
| - Dropout rate | 0.3 |
| | |
| **Model Architecture** | |
| - Input channels | 64 |
| - Sequence length | 2048 samples |
| - Embedding size | 96 |

