# OpenReview forum: "Learnable Fractional Fourier and Graph Fractional Operators for Nonstationary Graph Signals Validated with EEG Seizure Detection"
_ICLR.cc/2026/Conference — Submitted to ICLR 2026_

### Official Review · Reviewer_mqQh · 2025-10-29

**Soundness:** 2
**Presentation:** 2
**Contribution:** 2
**Rating:** 2
**Confidence:** 5

**Summary:**

This paper proposes EEG-GraphFrFT, a dual-path architecture combining a learnable fractional Fourier transform on raw EEG signals with a fractional graph operator applied to functional connectivity networks. The method aims to better handle non-stationary EEG and evolving graph topology in epileptic seizure detection.

**Strengths:**

Seizure detection using dynamic graph models is an important research topic.

The idea of jointly learning fractional orders in both temporal and graph domains is interesting.

**Weaknesses:**

**Unclear motivation and insufficient problem positioning.** The paper does not clearly articulate why seizure detection specifically requires a new fractional operator-based temporal GNN (T-GNN) architecture. The motivation provided is very generic (“standard deep models assume stationarity”), without demonstrating a concrete failure mode of existing seizure T-GNNs.

**Insufficient seizure-specific TGNN literature and baseline comparison.**
The related work omits a large body of domain-relevant work on dynamic seizure graph modeling (e.g., GCN-GRU [Tang et al, ICLR 2022], GRAPHS4MER [Tang et al, CHIL 2025],, EvoBrain (Rikuto et al, NeurIPS 2025), and other seizure T-GNN works. So, it is unclear what the technical novelty is relative to prior TGNNs for seizures, not just general GSP/Fourier methods.

So, the baselines are insufficient. Current comparisons primarily general-purpose time-series or transformer architectures (FreTS, iTransformer, Mamba). I believe there are large body of EEG baselines and popular foundation models (BIOT, LaBraM, or EEGgpt) should be considered. Also, this work does not include any TGNN-based seizure detection models, which makes it difficult to assess whether the proposed improvement comes from their design.

**Lacking seizure benchmark datasets.** The standard datasets used for seizure detection (TUSZ, CHB-MIT) should be included. I found that Helsinki is a neonatal dataset and carries different signal characteristics than adult seizure EEG, so it is unclear whether the results generalize. Conducting benchmarking experiments are basic to evaluate the performance of the proposal.

**Questions:**

Could you please illustrate the differences between the proposed method and existing T-GNNs?

---

### Official Review · Reviewer_SKCK · 2025-10-30

**Soundness:** 2
**Presentation:** 2
**Contribution:** 2
**Rating:** 2
**Confidence:** 3

**Summary:**

This paper propose EEG-GraphFrFT, a unified dual-path framework for nonstationary graph signals with time-varying spectral properties and evolveing network topologies with learnable fractional operators. Theoretical properties for stable training are established. The proposed framework is evaluated on three public datasets.

**Strengths:**

1. **Learnable fractional parameters** in both temporal and graph domains provide additional modeling flexibility beyond fixed spectral or fractional operators.

2. **Dual-path architecture with structured fusion** offers a clear mechanism to combine temporal frequency features with connectivity-based representations.

3. **Competitive performance and robustness**, demonstrated through strong results on EEG seizure tasks and stress tests under noise/channel perturbations.

**Weaknesses:**

**1. Motivation is not clear**

The paper argues that combining fractional temporal and graph operators addresses non-stationary EEG signals, whose amplitudes and frequencies evolve over time. However, it is not justified why classical time–frequency representations (e.g., multi-scale wavelet transform, which are explicitly designed to handle time-varying frequency content) are insufficient in this setting. Why the fractional transforms are preferable to established non-stationary analysis tools (e.g., wavelets, STFT variants, adaptive time–frequency bases)?

**2. Lack related work and connection to prior methods**

The related-work section briefly mentions FrFT-based and multi-scale wavelet approaches in computer vision, as well as graph fractional Fourier neural networks. However, the paper does not clearly articulate how these prior works relate to the proposed EEG-GraphFrFT model, nor how the presented method differs from them, weakening the significance of EEG-GraphFrFT model.

In addition, the discussion of fractional time–frequency methods is incomplete. Relevant recent work on fractional scattering and adaptive time–frequency representations (e.g., Zhao et al., IEEE TSP 2023; Chen et al., IEEE TGRS 2023) is not discussed, leaving a gap in positioning relative to established fractional and wavelet-based models.

[1] Zhao et al, Deep scattering network with fractional wavelet transform, IEEE TSP 2023.

[2] Chen et al, Ssn: stockwell scattering network for sar image change detection, IEEE TGRS 2023

**3. Trivial definition of the fractional pseudo-differential operator**

The defintion of Fractional Pseudo-Differential Operator $T_{\alpha}^{\theta}$ appears trival. It performs a windowed FrFT followed by an inverse transform, with a separable window in the time and fractional-frequency domains. Although the window $a(t,\xi)$ is stated to be learnable, the core formulation remains largely conventional. From a novelty/contribution perspective, it would be natural to expect the fractional order $\theta$ itself to be learnable within the framework; without this, the contribution of the proposed operator seems overstated.

**4. Mismatch between theoretical discussion and robustness evaluation**

The theoretical analysis focuses on the stability of graph fractional filters under structural perturbations of the Laplacian. However, the robustness experiments primarily evaluate performance under additive pink noise. A robustness study aligned with structural graph perturbations (e.g., edge noise, topology corruption) would better support the theoretical claims.

**5. Lack of complexity analysis**

The paper does not provide a computational complexity analysis for the proposed EEG-GraphFrFT model. It is unclear how the method scales with graph size and how training time, inference cost, and memory usage compare to standard spectral or graph-based baselines.

**Questions:**

1. For the Figure 1. Chanel 1 provides adaptive time-frequency analysis, and Chanel 2 offers the graph (spatical)-spectral analysis, how these outputs from these two Chanels could be fused and combined?


2.  In the ablation study of the learnable $\alpha$, the paper states that both channels use trainable $\alpha$. However, in the temporal branch (Channel 1), the operator is defined via the fractional Fourier transform, where the fractional order is typically denoted by $\theta$. It is unclear whether the $\alpha$ mentioned in the ablation refers to this FrFT order $\theta$, or to a different parameter. This notation inconsistency makes it difficult to interpret the ablation design.

3. According to the definition of the fractional order $\alpha$ in graph fractional filters, $\alpha$ is typically constrained to $(0,1]$. However, Table 4 reports values such as $\alpha=1.5$, why？

---

### Official Review · Reviewer_jGyp · 2025-10-31

**Soundness:** 2
**Presentation:** 2
**Contribution:** 2
**Rating:** 2
**Confidence:** 4

**Summary:**

EEG-GraphFrFT couples learnable fractional time–frequency analysis (FrFT) with graph-fractional filtering of dynamic functional networks—backed by stability guarantees and low-rank fusion—to achieve 2–8% accuracy gains and robust, subject-disjoint seizure detection, while generalizing to broader nonstationary graph signals.

**Strengths:**

1. The learnable fractional operators unify time-frequency and graph-space dynamics and come with stability (Hölder) and well-posedness results -rare for EEG deep models.\
2.  It claims consistent 2-8 % accuracy gains over state-of-the-art baselines in subject-disjoint settings, with resilience to noise/channel loss and a parameter-efficient fusion design.

**Weaknesses:**

1. Limited motivation for fractional modeling. The paper claims that fractional temporal–graph operators are essential for handling non-stationary EEG dynamics, yet it does not sufficiently differentiate the proposed approach from established time–frequency techniques (e.g., wavelets, adaptive TF representations). A more rigorous justification is needed to explain when and why fractional operators offer superior modeling capacity for EEG signals.

 2. Insufficient related works. While related work is briefly mentioned, the paper does not clearly position the EEG-GraphFrFT model relative to prior fractional or graph-based time–frequency methods. In particular, recent advances in fractional scattering and adaptive TF frameworks (e.g., Chen et al., IEEE TGRS 2025) are not discussed, leaving the novelty and methodological distinction under-explored.  Chen et al, Joint spatial-frequency scattering network for unsupervised SAR image change detection, IEEE TGRS, 2025.

 3.  No complexity analysis: The proposed model introduces fractional graph filters and windowed FrFT components, but no computational complexity or resource analysis is provided. It remains unclear how the method scales with network size and signal length, and how its training/inference cost compares with conventional graph spectral or TF-based baselines.

**Questions:**

See above weaknesses

---

### Meta-Review · Area_Chair_i7ir · 2026-01-05

**Summary:**

This paper introduces EEG-GraphFrFT, a dual-path framework built on learnable fractional operators that unify time-frequency analysis and functional connectivity. In the proposed framework, the temporal path uses a fractional Fourier transform with trainable order to capture transient, nonstationary patterns, while the graph path constructs networks from wPLI and Spectral Granger causality and applies fractional operators to model complex interactions. Validation on three public seizure datasets (FMCE, HUP, Helsinki neonatal EEG) under subject-disjoint conditions shows accuracy gains over baselines.

This paper initially receives three reject scores. The three reviewers had concerns about the innovation and contribution of this submission and these issues have not been solved, so it did not meet the acceptance criteria of ICLR.

**Reviewer Concerns:**

Based on the reviewers' comments, the paper suffers from the following issues:

1. unclear motivation for fractional modeling (#jGyp, #SKCK, #mqQh),

2. insufficient related work and poor positioning against prior fractional and seizure-specific TGNN methods (#jGyp, #SKCK, #mqQh),

3. weak novelty in operator design (#SKCK),

4. mismatch between theoretical claims and robustness experiments (#SKCK),

5. omission of standard seizure benchmarks and baselines (#mqQh).

The concerns raised by the reviewers were not responded to by the authors of the paper during the rebuttal stage.

**Reviewer Scores:**

The authors gave up the rebuttal, so there was no discussion between them and the reviewers would not have changed their score.

---

### Decision · Program_Chairs · 2026-01-26

Reject